# A genetic screen in macrophages identifies new regulators of IFNγ-inducible MHCII that contribute to T cell activation

Michael C Kiritsy[1†], Laurisa M Ankley[2†], Justin Trombley[2], Gabrielle P Huizinga[2], Audrey E Lord[1], Pontus Orning[3], Roland Elling[3], Katherine A Fitzgerald[3], Andrew J Olive[2]*

[1]Department of Microbiology and Physiological Systems, University of Massachusetts Medical School, Worcester, United States; [2]Department of Microbiology & Molecular Genetics, College of Osteopathic Medicine, Michigan State University, East Lansing, United States; [3]Division of Infectious Disease and Immunology, Department of Medicine, University of Massachusetts Medical School, Worcester, United States

*For correspondence:
oliveand@msu.edu

†These authors contributed equally to this work

Competing interest: The authors declare that no competing interests exist.

## Abstract

Cytokine-mediated activation of host immunity is central to the control of pathogens. Interferon-gamma (IFNγ) is a key cytokine in protective immunity that induces major histocompatibility complex class II molecules (MHCII) to amplify CD4+ T cell activation and effector function. Despite its central role, the dynamic regulation of IFNγ-induced MHCII is not well understood. Using a genome-wide CRISPR-Cas9 screen in murine macrophages, we identified genes that control MHCII surface expression. Mechanistic studies uncovered two parallel pathways of IFNγ-mediated MHCII control that require the multifunctional glycogen synthase kinase three beta (GSK3β) or the mediator complex subunit 16 (MED16). Both pathways control distinct aspects of the IFNγ response and are necessary for IFNγ-mediated induction of the MHCII transactivator *Ciita*, MHCII expression, and CD4+ T cell activation. Our results define previously unappreciated regulation of MHCII expression that is required to control CD4+ T cell responses.

## Editor's evaluation

In this study, Olive and colleagues used a genetic screen to identify new regulators underpinning the ability of the cytokine IFNγ to upregulate MHC class II molecules, of relevance to our understanding of how macrophages are activated by IFNγ to confer host defense during microbial infection. They identified the signaling protein GSK3β, and MED16, a subunit of the Mediator complex previously implicated in gene induction.

## Introduction

Activation of the host response to infection requires the coordinated interaction between antigen presenting cells (APCs) and T cells (*van Elsland and Neefjes, 2018*; *Iwasaki and Medzhitov, 2015*; *Tubo and Jenkins, 2014*). For CD4+ T cells, the binding of the T cell receptor (TCR) to the peptide-loaded major histocompatibility complex class II (MHCII) on the surface of APCs is necessary for both CD4+ T cell activation and their continued effector function in peripheral tissues (*Tubo and Jenkins, 2014*; *DeSandro et al., 1999*; *Reith et al., 2005*). Dysregulation of MHCII control leads to a variety of conditions including the development of autoimmunity and increased susceptibility to pathogens

and cancers (*Koyama et al., 2011*; *Abrahimi et al., 2016*; *Thelemann et al., 2016*; *Johnson et al., 2016*; *Steimle et al., 1993*). While MHCII is constitutively expressed on dendritic cells and B cells, the production of the cytokine IFNγ promotes MHCII expression broadly in other cellular populations including macrophages (*Jakubzick et al., 2017*; *Unanue et al., 2016*; *Collins et al., 1984*; *Neefjes et al., 2011*). The induction of MHCII in these tissues activates a feedforward loop wherein IFNγ-producing CD4[+] T cells induce myeloid MHCII expression, which in turn amplifies CD4[+] T cell responses (*Neefjes et al., 2011*; *Buxadé et al., 2018*; *Ivashkiv, 2018*). Thus, IFNγ-mediated MHCII expression is essential for protective immunity.

The IFNγ-dependent control of MHCII is complex (*Reith et al., 2005*; *Unanue et al., 2016*; *Wijdeven et al., 2018*; *Herrero et al., 2001*; *Ting and Trowsdale, 2002*). Binding of IFNγ to its receptor induces cytoskeletal and membrane rearrangement that results in the activation of JAK1 and JAK2 and STAT1-dependent transcription (*Bousoik and Montazeri Aliabadi, 2018*; *Hu and Ivashkiv, 2009*). STAT1 induces *Irf1*, which then drives the expression of the MHCII master regulator, *Ciita* (*Schroder et al., 2004*; *Lehtonen et al., 1997*). The activation of CIITA opens the chromatin environment surrounding the MHCII locus and recruits transcription factors, including CREB1 and RFX5 (*Reith et al., 2005*; *Beresford and Boss, 2001*). MHCII is also regulated post-translationally to control the trafficking, peptide loading, and stability of MHCII on the surface of cells (*Paul et al., 2011*; *Oh et al., 2013*; *Alix et al., 2020*). While recent evidence points to additional regulatory mechanisms of IFNγ-mediated MHCII expression, including the response to oxidative stress, these have not been investigated directly in macrophages (*Wijdeven et al., 2018*).

In non-inflammatory conditions, macrophages express low levels of MHCII that is uniquely dependent on NFAT5 (*Buxadé et al., 2018*). While basal MHCII expression on macrophages plays a role in graft rejection, it is insufficient to control intracellular bacterial pathogens, which require IFNγ-activation to propagate protective CD4[+] T cell responses (*Ankley et al., 2020*; *Grau et al., 1998*; *Underhill et al., 1999*). Many pathogens including *Mycobacterium tuberculosis* and *Chlamydia trachomatis* inhibit IFNγ-mediated MHCII induction to evade CD4[+] T-cell-mediated control and drive pathogen persistence (*Pai et al., 2003*; *Pennini et al., 2006*; *Zhong et al., 1999*). Overcoming these pathogen immune evasion tactics is essential to develop new treatments or immunization strategies that provide long-term protection (*Ankley et al., 2020*). Without a full understanding of the global mechanisms controlling IFNγ-mediated MHCII regulation in macrophages, it has proven difficult to dissect the mechanisms related to MHCII expression that cause disease or lead to infection susceptibility.

Here, we globally defined the regulatory networks that control IFNγ-mediated MHCII surface expression on macrophages. Using CRISPR-Cas9 to perform a forward genetic screen, we identified the major components of the IFNγ-regulatory pathway in addition to many genes with no previously known role in MHCII regulation. Follow-up studies identified two critical regulators of IFNγ-dependent *Ciita* expression in macrophages, MED16 and GSK3β. Loss of either MED16 or GSK3β resulted in significantly reduced MHCII expression on macrophages, unique changes in the IFNγ-transcriptional landscape, and prevented the effective activation of CD4[+] T cells. These results show that IFNγ-mediated MHCII expression in macrophages is finely tuned through parallel regulatory networks that interact to drive efficient CD4[+] T cell responses.

## Results

### Optimization of CRISPR-Cas9 editing in macrophages to identify regulators of IFNγ-inducible MHCII

To better understand the regulation of IFNγ-inducible MHCII, we optimized gene-editing in immortalized bone marrow-derived macrophages (iBMDMs) from C57BL/6 J mice. iBMDMs were transduced with Cas9-expessing lentivirus and Cas9-mediated editing was evaluated by targeting the surface protein CD11b with two distinct single guide RNAs (sgRNA). When we compared CD11b surface expression to a non-targeting control (NTC) sgRNA by flow cytometry, we observed less than 50 % of cells targeted with either of the *Cd11b* sgRNA were successfully edited (*Figure 1—figure supplement 1A*). We hypothesized that the polyclonal Cas9-iBMDM cells variably expressed Cas9 leading to inefficient editing. To address this, we isolated a clonal population of Cas9-iBMDMs using limiting dilution plating. Using the same *Cd11b* sgRNAs in a clonal population (clone L3) we found 85–99% of cells were deficient in CD11b expression by flow cytometry compared to NTC (*Figure 1—figure*

*supplement 1B*). Successful editing was verified by genotyping the *Cd11b* locus for indels at the sgRNA targeting sequence using Tracking of Indels by Decomposition (TIDE) analysis (*Brinkman et al., 2014*). Therefore, clone L3 Cas9+ iBMDMs proved to be a robust tool for gene editing in murine macrophages.

To test the suitability of these cells to dissect IFNγ-mediated MHCII induction, we next targeted *Rfx5*, a known regulator of MHCII expression, with two independent sgRNAs (*Steimle et al., 1995*). Since L3 macrophages do not express IFNγ, we stimulated *Rfx5* targeted and NTC cells with IFNγ for 18 hours and quantified the surface expression of MHCII by flow cytometry (*Figure 1A and B* and *Figure 5—source data 1*). In cells expressing the non-targeting sgRNA, IFNγ stimulation resulted in a 20-fold increase in MHCII. In contrast, cells transduced with either of two independent sgRNAs targeting *Rfx5* failed to induce the surface expression of MHCII following IFNγ stimulation. We further tested other activators that might impact MHCII expression in L3 cells. L3 cells were stimulated with IFNγ, LPS, Pam3CSK4, IFN-β, TNF and N-glycolylated muramyldipeptide (NG-MDP) and 24 hours later the surface expression of MHCII and PD-L1 was quantified. While each stimuli increased PD-L1 expression, only IFNγ significantly altered the expression of MHCII (*Figure 1—figure supplement 1C,D* ). Thus, MHCII expression in macrophages is tightly controlled by IFNγ-dependent mechanisms and L3 cells can be effectively used to interrogate IFNγ-mediated MHCII expression in macrophages.

## Forward genetic screen identifies known and novel regulators of MHCII surface expression in macrophages

To define the genetic networks required for IFNγ-mediated MHCII expression, we made a genome-wide library of mutant macrophages with sgRNAs from the Brie library to generate null alleles in all protein-coding genes (*Doench et al., 2016*). After verifying coverage and minimal skew in the initial library, we conducted a forward genetic screen to identify regulators of IFNγ-dependent MHCII expression (*Figure 1C* and *Supplementary file 1*). The loss-of-function library was stimulated with IFNγ and 24 hours later, we selected MHCII^high and MHCII^low expressing cells by fluorescence activated cells sorting (FACS). Following genomic DNA extraction, sgRNA abundances for each sorted bin were determined by deep sequencing.

As our knockout library relied on the formation of Cas9-induced indels and was exclusive to protein-coding genes, we focused our analysis on genes expressed in macrophages under the conditions of interest, which we determined empirically in the isogenic cell line by RNA-seq (*Figure 5—source data 1*). We assumed that sgRNAs targeting non-transcribed genes are neutral in their effect on IFNγ-induced MHCII expression, which afforded us ~32,000 internal negative control sgRNAs (*Hart et al., 2014*). To test for statistical enrichment of sgRNAs and genes, we used the modified robust rank algorithm (α-RRA) employed by Model-based Analysis of Genome-wide CRISPR/Cas9 Knockout (MAGeCK), which first ranks sgRNAs by effect and then filters low ranking sgRNAs to improve gene significance testing (*Li et al., 2014*). We tuned the sgRNA threshold parameter to optimize the number of significant hits without compromising the calculated q-values of known positive controls that are expected to be required for IFNγ-mediated MHCII expression. Further, by removing irrelevant sgRNAs that targeted genes not transcribed in our conditions, we removed potential false positives and improved the positive predictive value of the screen (*Figure 1—figure supplement 2A* and S2B).

Guide-level analysis confirmed the ability to detect positive control sgRNAs which had robust enrichment in the MHCII^low population (*Figure 1—figure supplement 2C*). Using the previously determined parameters, we tested for significantly enriched genes that regulated MHCII surface levels. As expected, sgRNAs targeting known components of the IFNγ-receptor signal transduction pathway, such as *Ifngr1*, *Ifngr2*, *Jak1* and *Stat1*, as well as regulators and components of IFNγ−mediated MHCII expression, such as *Ciita*, *Rfx5*, and *Rfxank* were all significantly enriched (*Figure 1D*; *Reith et al., 2005*; *Schroder et al., 2004*). These results validated our approach to identify functional regulators of IFNγ-mediated MHCII expression.

Stringent analysis revealed a significant enrichment of genes with no known involvement in interferon responses and antigen presentation. To identify functional pathways that are associated with these genes, we performed KEGG pathway analysis on the positive regulators of IFNγ-induced MHCII that met the FDR cutoff (*Figure 1—figure supplement 2D*; *Kanehisa and Goto, 2000*; *Kanehisa et al., 2019*; *Kanehisa, 2019*). However, gene membership for the 10 most enriched KEGG pathways was largely dominated by known regulators of IFNγ signaling. To circumvent this redundancy

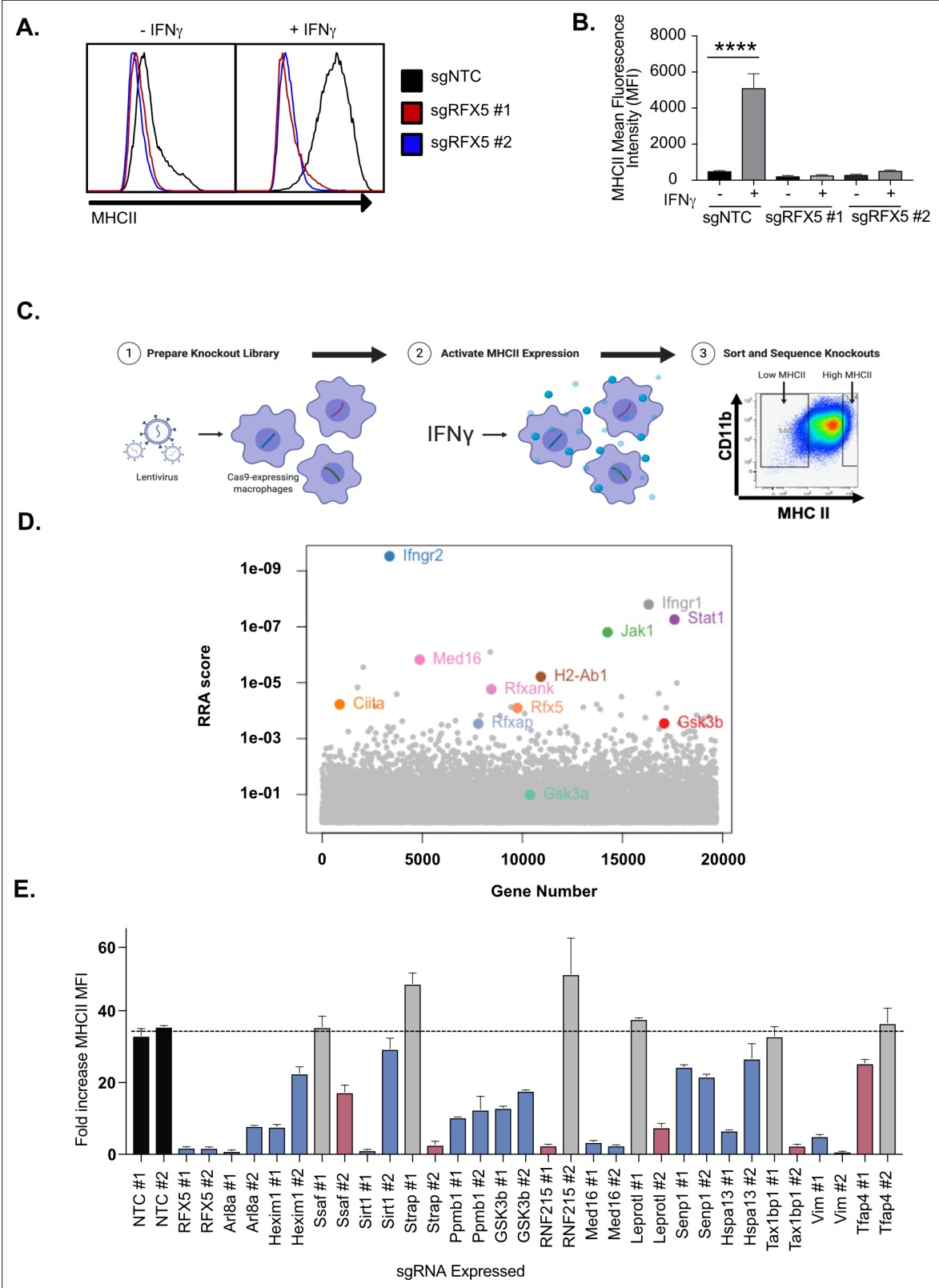

**Figure 1.** Genome-wide CRISPR Cas9 screen identifies regulators of IFN $\gamma$-dependent MHCII expression. (**A**) Cas9+ iBMDMs (Clone L3) expressing the indicated sgRNAs were left untreated or treated with IFN $\gamma$ (6.25 ng/ml) for 24 hours. Surface MHCII was quantified by flow cytometry. Shown is a representative histogram of MHCII surface staining and (**B**) the quantification of the mean fluorescence intensity (MFI) in the presence and absence of IFN $\gamma$ stimulation from three biological replicates. **** p < 0.0001 by one-way ANOVA with tukey correction for multiple hypotheses. These data

*Figure 1 continued on next page*

*Figure 1 continued*

are representative of three independent experiments. (**C**) A schematic representation of the CRISPR-Cas9 screen conducted to identify regulators of IFN γ -inducible MHCII surface expression on macrophages. A genome-wide CRISPR Cas9 library was generated in L3 cells using sgRNAs from the Brie library (four sgRNAs per gene). The library was treated with IFN γ and MHCII^hi and MHCII^low populations were isolated by FACS. The representation of sgRNAs in each population in addition to input library were sequenced. (**D**) Shown is score for each gene in the CRISPR-Cas9 library that passed filtering metrics as determined by the alpha-robust rank algorithm (a-RRA) in MAGeCK from two independent screen replicates. (**E**) The L3 clone was transduced with the indicated sgRNAs for candidates (two per candidate gene) in the top 100 candidates from the CRISPR-Cas9 screen. All cells were left untreated or treated with 10 ng/µl of IFN γ for 24 hours then were analyzed by flow cytometry. The fold-increase in MFI was calculated for triplicate samples for each cell line (MFI IFN γ +/MFI IFN γ -). The results are representative of at least two independent experiments. Candidates that were significant for two sgRNAs (Red) or one sgRNA (Blue) by one-way ANOVA compared to the mean of NTC1 and NTC2 using Dunnets multiple comparison test. Non-significant results are shown in gray bars.

The online version of this article includes the following figure supplement(s) for figure 1:

**Source data 1.** CRISPR screen analysis.

**Figure supplement 1.** Optimization of CRISPR-Cas9 editing in iBMDMs.

**Figure supplement 2.** Adaptations to the MAGeCK analysis pipeline identifies high confidence regulators of IFN γ -mediated MHCII expression following a Genome-wide CRISPR Cas9 screen.

and identify novel pathways enriched from our candidate gene list, the gene list was truncated to remove the 11 known IFNγ signaling regulators. Upon reanalysis, several novel pathways emerged, including mTOR signaling (*Figure 1—figure supplement 2E*). Thus, our genetic screen uncovered previously undescribed pathways that are critical to control IFNγ-mediated MHCII surface expression in macrophages.

The results of the genome-wide CRISPR screen highlight the sensitivity and specificity of our approach and analysis pipeline. To gain new insights into IFNγ-mediated MHCII regulation, we next validated a subset of candidates that were not previously associated with the IFNγ-signaling pathway. Using two independent sgRNAs for each of 15 candidate genes, we generated loss-of-function macrophages in the L3 clone. MHCII surface expression was quantified by flow cytometry for each cell line in the presence and absence of IFNγ activation. For all 15 candidates, we observed no changes in basal MHCII expression (*Figure 1—figure supplement 2F*) but found deficient MHCII induction following IFNγ stimulation with at least one sgRNA (*Figure 1E* and *Figure 1—figure supplement 2G*). For 9 of 15 candidate genes, we observed a significant reduction in MHCII surface expression with both gene-specific sgRNAs These results show that our screen not only identified known regulators of IFNγ-mediated MHCII induction, but also uncovered new regulatory networks required for MHCII expression on macrophages.

We were interested in better understanding the IFNγ-mediated transcriptional activation of MHCII to determine if a subset of candidates reveal new regulatory mechanisms of MHCII-expression. Based on the screen and validation results, we examined the known functions of the candidates that were confirmed with two sgRNAs, and identified *Med16* and *Gsk3b* for follow-up study. MED16 is a subunit of the mediator complex that regulates transcription initiation while Glycogen synthase kinase 3β (GSK3β) is a multifunctional kinase that controls signaling pathways known to regulate transcription (*Poss et al., 2013*; *Wu and Pan, 2010*). Thus, we hypothesized that MED16 and GSK3β would be required for effective IFNγ-mediated transcriptional control of MHCII.

## MED16 is uniquely required for IFNγ-mediated CIITA expression

We first examined the role of MED16 in controlling IFNγ-mediated MHCII expression. Our valida-tion results confirmed that MED16 was indeed an essential positive regulator of MHCII expression (*Figure 1E*). MED16 was the sixth ranked candidate from our screen results, with robust enrichment of all four sgRNAs in the MHCII^low population (*Figure 2A*). As part of the mediator complex, MED16 bridges the transcription factor binding and the chromatin remodeling that are required for transcrip-tional activation (*Conaway and Conaway, 2011*). These changes then recruit and activate RNA poly-merase II to initiate transcription. While the core mediator complex function is required for many RNA polymerase II dependent transcripts, distinct sub-units of the mediator complex can also play unique roles in gene regulation (*Poss et al., 2013*; *Conaway and Conaway, 2011*). To examine if MED16 was uniquely required for IFNγ-dependent MHCII expression, we probed our genetic screen data for all mediator complex subunits. The other 27 mediator complex subunits in our library did not show

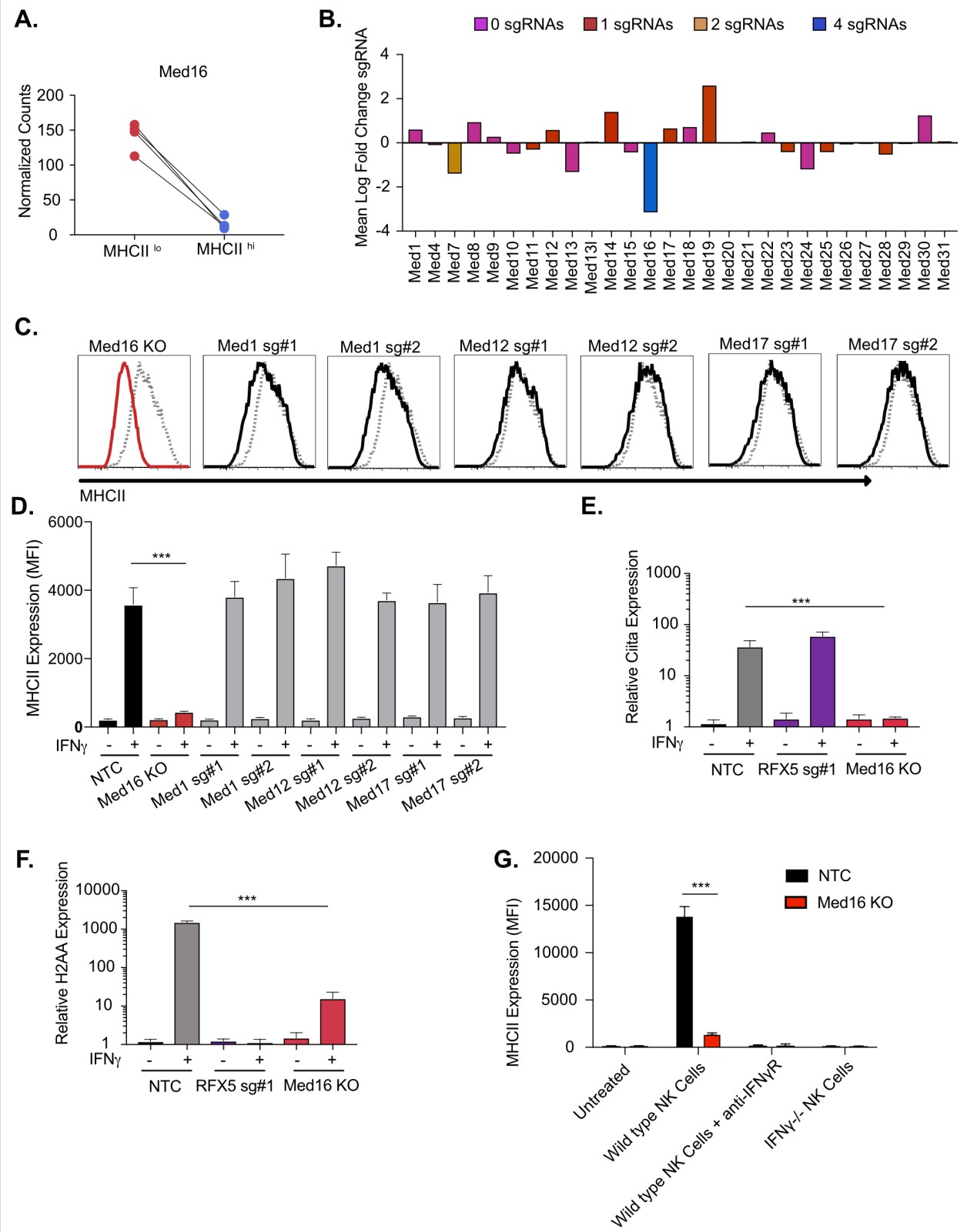

**Figure 2.** The mediator complex subunit MED16 is uniquely required for IFN $\gamma$ -mediated MHCII surface expression. (**A**) Shown is the normalized mean read counts from FACS sorted MHCII^low and MHCII^hi populations for the four sgRNAs targeting *Med16* within the genome-wide CRISPR-Cas9 library. (**B**) The mean of the log fold change (normalized counts in MHCII^hi/normalized counts in MHCII^low) for each mediator complex subunit that passed quality control metrics described in Materials and methods. The bar colors indicate the number of sgRNAs out of four possible that pass the alpha cutoff

*Figure 2 continued on next page*

*Figure 2 continued*

using the MAGeCK analysis pipeline as described in material and methods. (**C**) *Med16* KO cells or L3 cells targeted with the indicated sgRNA were left untreated or were treated with 6.25 ng/ml of IFN $\gamma$ for 18 hours. Cells were then analyzed for surface MHCII expression by flow cytometry. Shown are representative comparing the MHCII surface expression of indicated mediator complex subunit (Black solid line) treated with IFN $\gamma$ overlayed with NTC (Gray-dashed line) treated with IFN $\gamma$ . (**D**) Quantification of the MFI of surface MHCII from the experiment in (**C**) from three biological replicates. These results are representative of two independent experiments. (**E**) NTC L3 cells, RFX5 sg#1 cells, and *Med16* KO cells were left untreated or were treated with 6.25 ng/ml of IFN $\gamma$ . 18 hours later cells RNA was isolated and qRT-PCR was used to determine the relative expression of *Ciita* and (**F**) *H-2aa* compared to GAPDH controls from three biological replicates. (**G**) NK cells from wild type or IFN $\gamma$ -/- mice were activated with IL12/IL18 overnight then added to NTC or *Med16* KO cells in the presence or absence of IFN $\gamma$ R blocking antibody. Twenty-four hours later MHCII expression on macrophages was quantified by flow cytometry. The results are representative of three independent experiments. ***p < 0.001 as determined one-way ANOVA compared to NTC cells with a Dunnets test.

The online version of this article includes the following figure supplement(s) for figure 2:

**Figure supplement 1.** *Med16* KO cells are deficient in MHCII expression of a range of IFNγ concentrations and time points.

any significant changes in MHCII expression (*Figure 2B*). To test the specific requirement of MED16, we generated knockout macrophages in *Med16* (*Med16* KO) using two independent sgRNAs and targeted three additional mediator complex subunits, *Med1*, *Med12* and *Med17*. We treated with IFNγ and quantified the surface levels of MHCII by flow cytometry. In support of the screen results, *Med1*, *Med12* and *Med17* showed similar MHCII upregulation compared to NTC cells, while *Med16* targeted cells demonstrated defects in MHCII surface expression (*Figure 2C and D*). These results suggest that there is specificity to the requirement for MED16-dependent control of IFNγ-induced *Ciita* that is unique among the mediator complex subunits.

To understand the mechanisms of how MED16 regulates MHCII-induction, we assessed the transcriptional induction of MHCII in *Med16* KO cells. In macrophages, the IFNγ-mediated transcriptional induction of MHCII subunits requires the activation of CIITA that then, in complex with other factors like RFX5, initiates transcription at the MHCII locus (*Neefjes et al., 2011*; *Wijdeven et al., 2018*). To determine whether MED16 controls the transcriptional induction of MHCII, we stimulated NTC, *Med16* KO and *Rfx5* targeted cells with IFNγ for 18 hours and isolated RNA. Using qRT-PCR, we observed that loss of RFX5 did not impact the induction of *Ciita*, but had a profound defect in the expression of *H2aa* compared to NTC cells (*Figure 2E and F*). Loss of MED16 significantly inhibited the induction of both *Ciita* and *H2aa*. We further compared MHCII expression between NTC and *Med16* KO cells over time and with varying IFNγ concentrations observing robust inhibition of MHCII expression in all conditions (*Figure 2—figure supplement 1B-D*).

To ensure that the IFNγ treatments reflect physiological conditions, we developed a co-culture assay with macrophages and activated Natural Killer (NK) cells that produce IFNγ. NTC and *Med16* KO cells were left untreated or were incubated with activated NK cells for 18 hours then MHCII expression on the surface of the macrophages was quantified by flow cytometry (*Figure 2G*). In this model, induction of MHCII on macrophages was entirely dependent on NK cell-derived IFNγ as antibody-mediated blockade of IFNγ signaling or co-culture with IFNγ-/- NK cells did not significantly change macrophage surface expression of MHCII. While co-culture of NTC macrophages with wild type NK cells robustly induced MHCII on the surface, *Med16* KO macrophages had significantly reduced MHCII expression. Altogether these data suggest that MED16 controls the IFNγ-mediated induction of MHCII through upstream regulation of CIITA.

## GSK3 regulates the IFNγ-dependent induction of CIITA

We next examined the mechanisms of GSK3β control of IFNγ-mediated MHCII expression in more detail. GSK3β is involved in many cellular pathways, yet no role in regulating IFNγ-mediated MHCII expression has previously been described (*Wu and Pan, 2010*; *Beurel et al., 2010*; *Thomson et al., 2009*; *Xu et al., 2008*). *Gsk3b* was highly ranked in the screen showing strong effects of multiple sgRNAs (*Figure 3A*; *Thomson et al., 2009*). Our validation studies further showed that GSK3β is required for the effective induction of IFNγ-dependent MHCII (*Figure 1E*). To begin to understand the mechanisms controlling GSK3β-dependent regulation of MHCII expression, we generated *Gsk3b* knockout cells (*Gsk3b* KO) and verified that the loss of *Gsk3b* inhibited IFNγ-mediated MHCII surface expression (*Figure 3B* and *Figure 3—figure supplement 1A*). We next examined if the IFNγ-mediated transcriptional induction of *Ciita* or *H2aa* were reduced in *Gsk3b* KO cells. Loss of *Gsk3b*

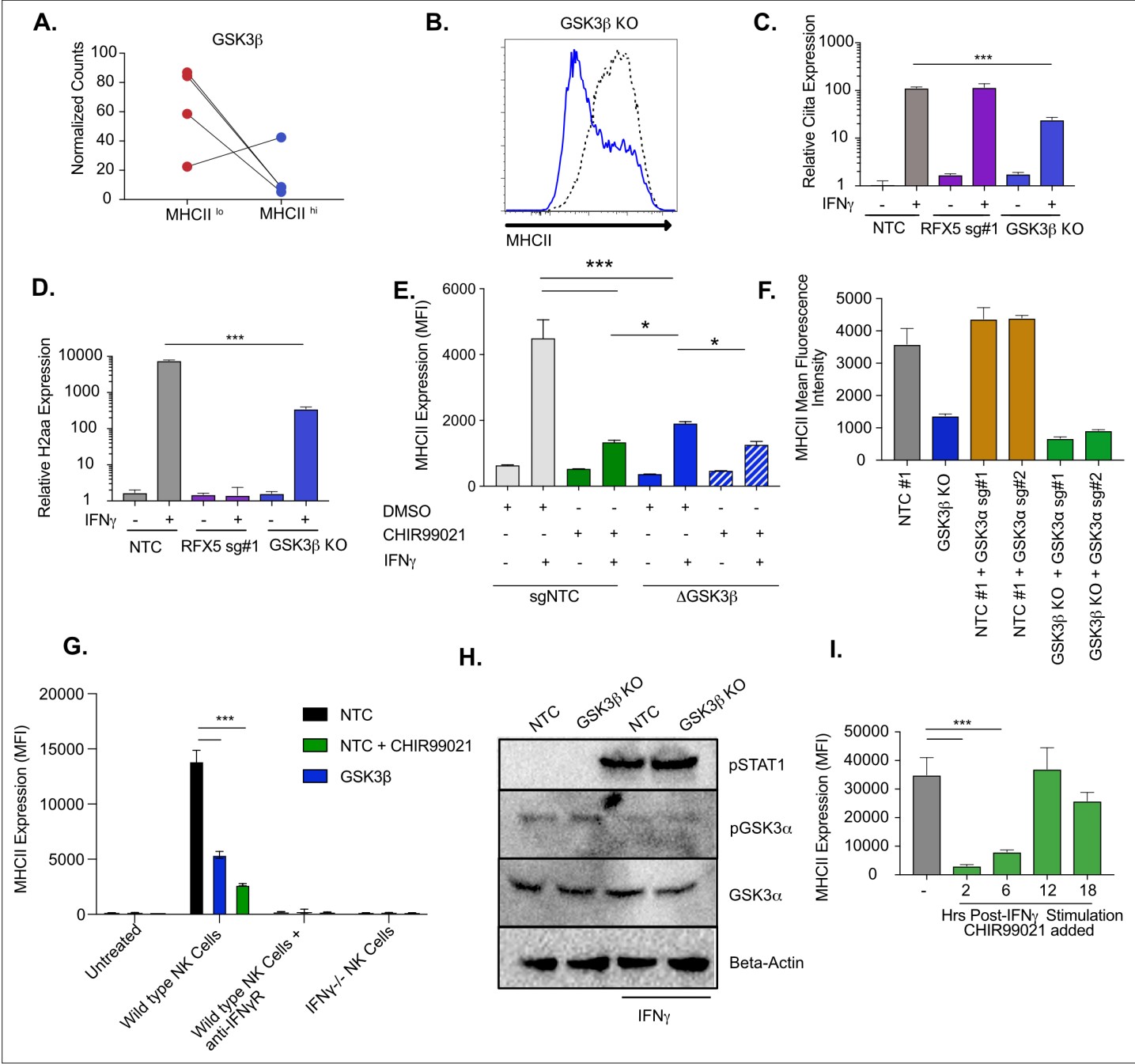

**Figure 3.** GSK3 $\beta$ and GSK3α coordinate IFN $\gamma$ -mediated CIITA and MHCII expression. (**A**) Shown is the normalized mean read counts from FACS sorted MHCII^low and MHCII^high populations for the four sgRNAs targeting *Gsk3b* within the genome-wide CRISPR-Cas9 library. (**B**) NTC L3 cells and *Gsk3b* KO cells were treated with 6.25 ng/ml of IFN $\gamma$ . Eighteen hr later, cells were stained for surface MHCII and analyzed by flow cytometry. Shown is a representative flow cytometry plot overlaying *Gsk3b* KO (blue line) with NTC (grey line). The results are representative of five independent experiments. (**C**) NTC L3 cells, *Rfx5* sg#1 cells, and *Gsk3b* KO cells were left untreated or were treated with 6.25 ng/ml of IFN $\gamma$ . Eighteen hr later, cells RNA was isolated and qRT-PCR was used to determine the relative expression of *Ciita* and (**D**) *H2aa* compared to *Gapdh* controls from three biological replicates. The results are representative of three independent experiments. (**E**) NTC L3 cells or Gsk3 $\beta$ KO were treated with DMSO or 10 μM CHIR99021 as indicated then left untreated or stimulated with IFN $\gamma$ for 18 hr. MHCII surface expression was then quantified by flow cytometry. The mean fluorescence intensity was quantified from three biological replicates. These results are representative of three independent experiments. (**F**) L3 cells or *Gsk3b* KO transduced with the indicated sgRNAs were treated with IFN $\gamma$ and 18 hr later the surface levels of MHCII were quantified by flow cytometry. The mean fluorescence intensity of surface MHCII was quantified from three biological replicates from this experiment that is representative of 4 independent experiments. (**G**) NK cells from wild type or IFN $\gamma$ -/- mice were activated with IL12/IL18 overnight then added to NTC or *Gsk3b* KO cells in the presence or absence of IFN $\gamma$ R blocking antibody, 10 μM CHIR99021 or DMSO. Twenty-four hours later, MHCII expression on macrophages was quantified by

*Figure 3 continued on next page*

*Figure 3 continued*

flow cytometry from three biological replicates. The results are representative of three independent experiments. (**H**) NTC or *Gsk3b* KO cells were left untreated or were stimulated with 6.25 ng/ml IFN $\gamma$ for 30 min. Cell lysates were used for immunoblot analysis with the indicated antibodies for pSTAT1, total GSK3α, pGSK3α, and Beta-actin. (**J**) Immortalized bone marrow macrophages were treated with IFN $\gamma$. Control cells were treated with DMSO and for the remaining cells CHIR999021 was added at the indicated times following IFN $\gamma$ treatment. 24 hours after IFN $\gamma$ stimulation the levels of surface MHCII were quantified by flow cytometry. Shown is the MFI for biological triplicate samples. ***p < 0.001 **p < 0.01 *p < 0.05 by one-way ANOVA with a Tukey Correction test.

The online version of this article includes the following figure supplement(s) for figure 3:

**Source data 1.** Raw Blots.

**Source data 2.** Labeled Blots.

**Figure supplement 1.** Gsk3 $\beta$ and GSK3α coordinate MHCII expression in macrophages.

significantly inhibited the expression of both CIITA and H2-Aa after IFNγ-treatment compared to NTC controls (*Figure 3C and D*). These data suggest that GSK3β, similar to MED16, is an upstream regulator of IFNγ-mediated MHCII induction and controls the expression of CIITA following IFNγ-activation. As with the *Med16* KO, we further compared MHCII expression between NTC and *Gsk3b* KO macrophages over time and with varying IFNγ concentrations observing significant inhibition of MHCII expression in all conditions (*Figure 3—figure supplement 1B-D*).

To confirm the genetic evidence using an orthogonal method, we next used the well-characterized small molecule CHIR99021, which inhibits both GSK3β and the GSK3β paralog GSK3α (*Wf et al., 2010*; *Ring et al., 2003*). NTC macrophages were treated with CHIR99021 and cells were then stimulated with IFNγ, and MHCII expression was quantified by flow cytometry. Inhibition of GSK3α/β activity reduced the induction of surface MHCII and was more deleterious than genetic loss of *Gsk3β* alone (*Figure 3E*). These data suggest a possible role for GSK3α in controlling IFNγ-mediated MHCII expression (*Huang et al., 2017*). While we did not observe enrichment for GSK3α in the screen (*Figure 2—figure supplement 1D* and *Supplementary file 1*), we could not exclude the possibility that GSK3α plays a key regulatory role during IFNγ activation when GSK3β is dysfunctional. We hypothesized that GSK3α can partially compensate for total loss of *Gsk3b*, resulting in some remaining IFNγ-induced MHCII expression. To test this hypothesis, we treated *Gsk3b* KO macrophages with CHIR99021 or DMSO and quantified MHCII surface expression. In support of an important regulatory role for GSK3α, CHIR99021 treatment of *Gsk3b* KO macrophages further reduced surface MHCII expression after IFNγ-stimulation compared to the *Gsk3b* KO alone (*Figure 3E*).

To exclude the possibility of CHIR99021 off-target effects we next targeted *Gsk3a* genetically. To enable positive selection of a second sgRNA, we engineered vectors in the sgOpti background with distinct resistance markers for bacterial and mammalian selection that facilitated multiplexed sgRNA cloning (see materials and methods) (*Fulco et al., 2016*). These vectors could be used to improve knockout efficiency when targeting a gene with multiple sgRNAs or target multiple genes simultaneously (*Figure 3—figure supplement 1E*). We targeted *Gsk3a* with two unique sgRNAs in either NTC or *Gsk3b* KO macrophages and stimulated the cells with IFNγ. Cells with the sgRNA targeting *Gsk3a* alone upregulated MHCII expression similarly to NTC control cells (*Figure 3F* and *Figure 3—figure supplement 1F*). In contrast, targeting *Gsk3a* in *Gsk3b* KO macrophages (i.e. double knockout) led to a further reduction of MHCII surface expression, similar to what was observed with CHIR99021 treatment. This same trend was observed when we examined *Ciita* mRNA expression after IFNγ-activation (*Figure 2—figure supplement 1G*). To ensure physiological levels of IFNγ, we next repeated the NK cell co-culture experiment with *Gsk3b* KO and CHIR99021 treated cells. We observed over a 3-fold reduction in MHCII expression in both conditions compared to NTC cells and the reduction was greater in CHIR99021 treated cells compared to *Gsk3b* KO cells (*Figure 3G*). As observed before, the MHCII induction was dependent on IFNγ as blocking the IFNγR with antibodies or co-culturing with IFNγ-/- NK cells resulted in no change in MHCII expression compared to no co-culture controls. Therefore, both GSK3β and GSK3α have important regulatory functions that control IFNγ-mediated MHCII expression.

We next examined possible mechanisms by which GSK3α controls MHCII expression only in the absence of GSK3β. We hypothesized that *Gsk3a* expression or activation is increased in the absence of GSK3β. To test these hypotheses, NTC and *Gsk3b* KO cells were left untreated or stimulated with IFNγ for 30 min. We measured total and phosphorylated GSK3α by immunoblot and observed

no significant difference between resting and IFNγ activation NTC and *Gsk3b* KO macrophages (*Figure 3H*). We observed robust phosphorylation of STAT1 further suggesting this pathway remains intact even in the absence of GSK3β. Together these data suggests that GSK3α does not compensate for the loss of GSK3β by modulating its expression or activation.

To understand the kinetics of the GSK3α/β requirement for IFNγ responses, we conducted a time course experiment with CHIR99021. We hypothesized that GSK3α/β inhibition with CHIR99021 would block MHCII expression only if the inhibitor was present shortly after IFNγ stimulation. To test this hypothesis, iBMDMs were stimulated with IFNγ then treated with DMSO for the length of the experiment or with CHIR99021, 2, 6, 12, and 18 hours post-stimulation. When MHCII was quantified by flow cytometry we saw a reduction in MHCII expression when CHIR99021 was added 2 or 6 hours after IFNγ (*Figure 3I*). CHIR99021 addition at later time points resulted in similar MHCII expression compared to DMSO treated cells. When the expression of *H2aa* mRNA was quantified from a parallel experiment, a significant reduction in mRNA expression was only observed in macrophages that were treated with CHIR99021 2 hours following IFNγ-activation (*Figure 2—figure supplement 1H*). Thus, GSK3α/β activity is required early after IFNγ stimulation to activate the transcription of MHCII. We repeated this experiment in primary bone marrow-derived macrophages from HoxB8 conditionally immortalized progenitor cells and observed comparable results (*Figure 2—figure supplement 1I*; *Wang et al., 2006*). Therefore, GSK3α/β activity is required for the effective induction of IFNγ-mediated MHCII in immortalized and primary murine macrophages and has a negligible effect on the maintenance or stability of cell surface-associated MHCII.

## GSK3α/β and MED16 function independently from mTORC1 to control IFNγ-mediated MHCII expression

Since the loss of either MED16 or GSK3β reduced IFNγ-mediated CIITA transcription, it remained possible that these two genes control MHCII expression through the same regulatory pathway. While *Med16* KO macrophages are greatly reduced in IFNγ-mediated MHCII induction, there remains a small yet reproducible increase in MHCII surface expression. We determined if this effect on MHCII expression after IFNγ-activation required GSK3 activity by treating *Med16* KO and NTC macrophages with CHIR99021. While DMSO-treated *Med16* KO cells showed a reproducible two- to threefold increase in MHCII expression after IFNγ stimulation, CHIR99021 treated *Med16* KO cells showed no change whatsoever (*Figure 4A*). CHIR99021 treatment of NTC cells resulted in a significant reduction in MHCII compared to vehicle controls. However, we observed more MHCII expression compared to CHIR99021 treated *Med16* KO cells. These results suggest that MED16 and GSK3α/β control IFNγ-mediated *Ciita* induction and MHCII expression through independent mechanisms.

Our bioinformatic analysis identified an enrichment for the mTOR pathway among positive regulators of MHCII expression. In contrast, a previous study linked IFNγ activation in human monocyte derived macrophages with the inhibition of mTORC1 (*Su et al., 2015*). Given this inconsistency and the previously described role of mTORC1 modulating GSK3 activity, we next examined how mTORC1 contributes to IFNγ-mediated MHCII expression. As a first step, we tested how the inhibition of mTORC1 impacts IFNγ responses in murine macrophages. NTC macrophages were treated with and without the mTORC1 inhibitor Torin2 then were left untreated or were stimulated with IFNγ. The surface expression of MHCII was then quantified by flow cytometry. While Torin2 alone had no effect on MHCII expression, blocking mTORC1 resulted in a significant reduction in surface MHCII following IFNγ activation, consistent with our screen analysis (*Figure 4C*). To determine the specificity of mTORC1 inhibition on other IFNγ responses we also examined the induction of the immunoinhibitory molecule programmed death ligand 1 (PD-L1) (*Figure 4D*). In contrast to MHCII, blockade of mTORC1 resulted in a significant increase in IFNγ-dependent PD-L1 expression compared to vehicle controls. Thus, the expression of distinct IFNγ-mediated genes are differentially controlled by mTOR signaling.

Since blocking mTORC1 inhibited IFNγ-mediated MHCII expression, we next tested whether mTORC1 functions in the same pathway as GSK3α/β or MED16. NTC cells with and without the inhibitor CHIR99021 in addition to *Gsk3b* KO and *Med16* KO macrophages were treated with low and high concentrations of Torin2. These cells were then activated with IFNγ and the surface expression of MHCII and PD-L1 was quantified by flow cytometry 24 hours later (*Figure 4D and E*). Consistent with our findings above, for all genotypes and treatments the inhibition of mTORC1 resulted in a

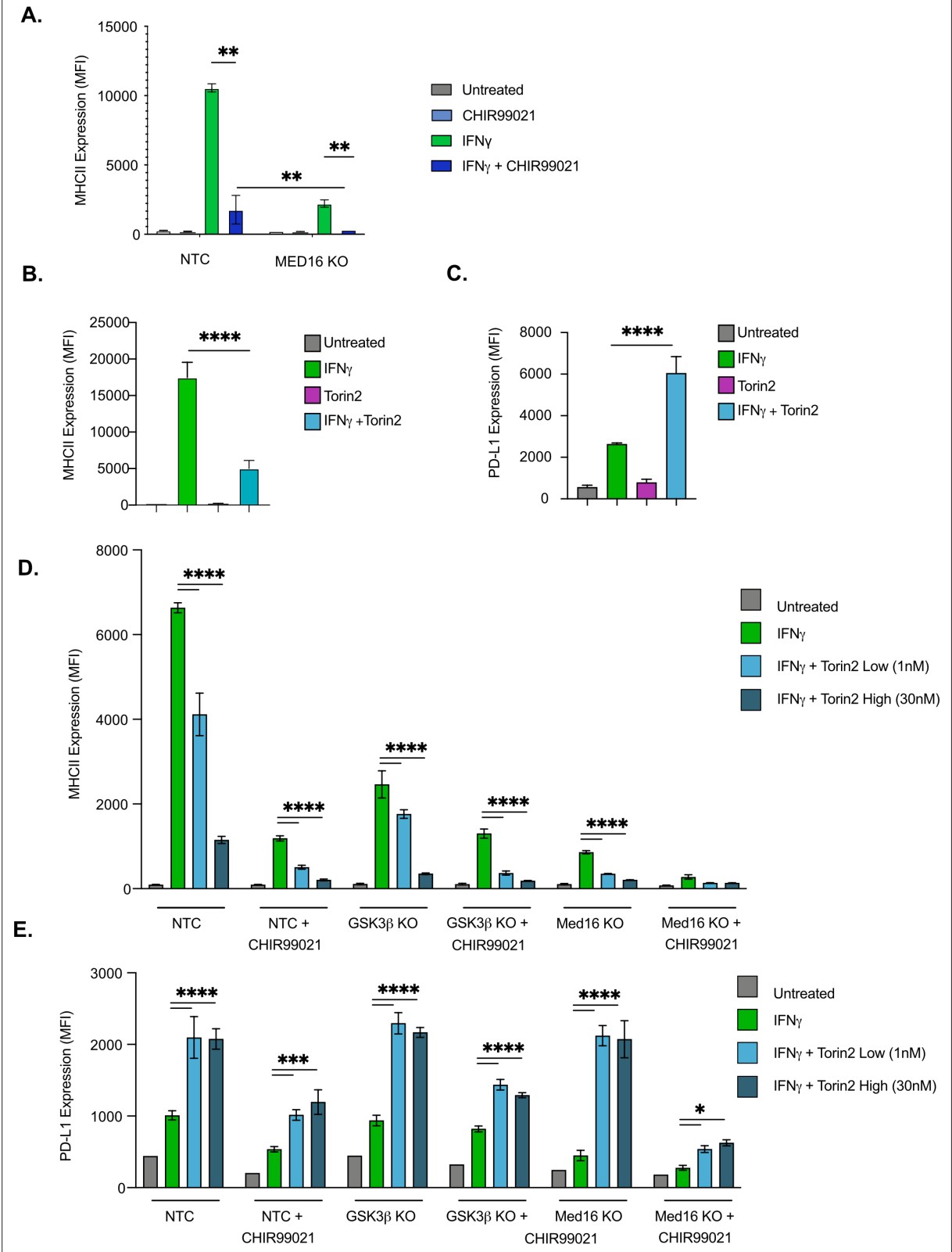

**Figure 4.** GSK3α/β and Med16 function independently from mTORC1 to control IFNγ-mediated MHCII expression. (**A**) NTC or *Med16* KO cells were treated with DMSO or CHIR99021 then left untreated or stimulated with IFNγ overnight. The following day MHC II cell surface expression was determined by flow cytometry. The quantification of the MFI of MHCII from four biological replicates is shown. **p < 0.001 by two-way ANOVA with multiple comparison correction. (**B and C**) NTC cells were treated with DMSO or 30 nM Torin2 for 2-hr then were stimulated with 6.25 ng/ml IFNγ

*Figure 4 continued on next page*

Figure 4 continued
overnight. Eighteen hr later (**B**) MHCII expression and (**C**) PD-L1 expression were quantified by flow cytometry. Shown is the MFI of the indicated marker from three biological replicates and is representative of three independent experiments. (**D and E**) NTC, *Gsk3b* KO and *Med16* KO cells were treated with DMSO or 10 uM CHIR99021 and/or the indicated Torin2 for 2 hours. Cells were then treated with IFN γ and the surface expression of (**D**) MHCII and (**E**) PD-L1 were quantified by flow cytometry. Shown is the MFI of the indicated marker from three biological replicates and is representative of three independent experiments. *p < 0.05, **p < 0.01, ***p < 0.001, **** p < 0.0001 by one or two-way ANOVA with correction for multiple comparisons.

significant reduction in MHCII expression and a significant increase in PD-L1. Taken together these data suggest that while mTORC1 is required for robust IFNγ-mediated MHCII expression, it functions independently of Med16 and GSK3α/β.

## GSK3β and MED16 control the expression of distinct IFNγ-mediated genes in macrophages

While GSK3β and MED16 independently regulate MHCII expression, their overlap in transcriptional regulation globally remained unknown. To test this, we compared the transcriptional profiles of *Med16* KO and *Gsk3b* KO cells to NTC cells by performing RNAseq on cells that were left untreated or were stimulated with IFNγ (See materials and methods). Principal component analysis of these six transcriptomes revealed distinct effects of IFNγ-stimulation ('condition'; PC1) and genotype (PC2) gene expression (*Figure 5A*). Both *Med16* and *Gsk3b* knockout macrophages had distinct transcriptional signatures in the absence of cytokine stimulation, which were further differentiated with IFNγ-stimulation. The PCA analysis suggested that MED16 and GSK3β control distinct transcriptional networks in macrophages following IFNγ-activation.

Transcriptional analysis confirmed a critical role of GSK3β and MED16 in regulating IFNγ-dependent *Ciita* and MHCII expression in macrophages compared to NTC controls (*Figure 5B and C*). However, the extent to which MED16 or GSK3β controlled the overall response of macrophages to IFNγ remained unclear. To directly assess how MED16 and GSK3β regulate the general response to IFNγ, we queried IFNγ-regulated genes from our dataset that are annotated as part of the cellular response to IFNγ stimulation (GeneOntology:0071346). Hierarchical clustering found that, of the 20 most induced IFNγ-regulated transcripts, the expression of eight were unaffected by loss of either *Gsk3b* and *Med16* (*Figure 5D*, Cluster 2). Importantly, these genes included a major regulator of the IFNγ response, *Irf1*, as well as canonical STAT1-target genes (*Gbp2, Gbp3, Gbp5, Gbp6 and Gbp7*). This suggests that neither GSK3β nor MED16 are global regulators of the IFNγ response in macrophages, but rather are likely to exert their effect on particular genes at the level of transcription or further downstream. In contrast, only two genes, out of the top 20 IFNγ-regulated genes, were similarly reduced in both *Med16* KO and *Gsk3b* KO cells (Cluster 4), one of which was *H2ab1*. This shows that while GSK3β and MED16 both regulate IFNγ-mediated MHCII expression, they otherwise control distinct aspects of the IFNγ-mediated response in macrophages. The remaining clusters from this analysis showed specific changes in either *Med16* KO or *Gsk3b* KO cells. Clusters 1 and 3 showed a subset of genes that were more robustly induced in *Gsk3b* KO cells compared to NTC and *Med16* KO cells. These genes included *Nos2, Il12rb1* and chemokines *Ccl2, Ccl3, Ccl4,* and *Ccl7*. In contrast, Cluster five showed a subset of genes that were reduced only in macrophages lacking MED16, including *Irf8* and *Stat1*; as these effects were modest, and did not reach statistical significance, they may be suggestive of an incomplete positive feedforward in which MED16 plays a role. Further stringent differential gene expression analysis (FDR < 0.05, absolute LFC > 1) of the IFNγ-stimulated transcriptomes identified 69 and 90 significantly different genes for MED16 and GSK3β respectively. Of these differentially expressed genes (DEGs), eight non-MHCII genes were shared between MED16 and GSK3β, including five genes that are involved in controlling the extracellular matrix (*Mmmp8, Mmp12, Tnn,* and *Clec12a*). Taken together these results suggest that while MED16 and GSK3β both regulate IFNγ-mediated *Ciita* and MHCII expression in macrophages, they otherwise control distinct regulatory networks in response to IFNγ.

We next used the transcriptional dataset to understand what aspects of IFNγ-mediated signaling MED16 and GSK3β specifically control. To resolve the transcriptional landscape of *Med16* KO macrophages and to understand the specific effect that MED16 loss has on the host response to IFNγ, we analyzed the DEGs for upstream regulators whose effects would explain the observed gene expression signature. The analysis correctly predicted a relative inhibition on IFNγ signaling compared

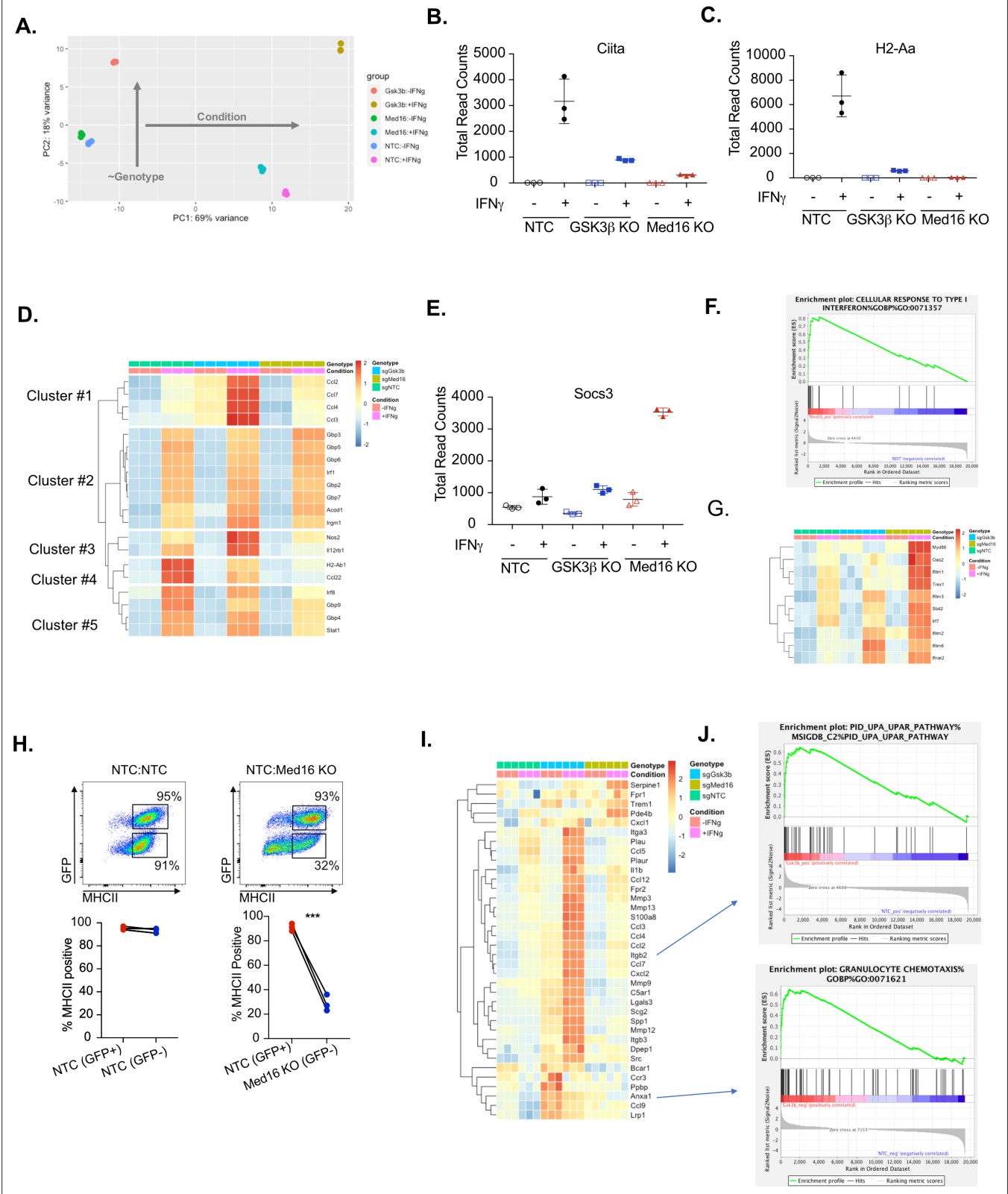

**Figure 5.** Transcriptomic analysis reveals distinct regulatory mechanisms of IFN γ signaling mediated by MED16 and GSK3 β. (**A**) The Global transcriptomes of NTC, *Gsk3b* KO and *Med16* KO was determined in the presence and absence of IFN γ -stimulation for 18 hours by RNA sequencing. Shown is the principal component analysis of the transcriptomes from three biological replicates for each condition. Dotplot showing the normalized read counts for (**B**) CIITA and (**C**) H2-Aa. (**D**) Shown is a heatmap showing the relative expression (log normalized, row-scaled) of the most varied 20

*Figure 5 continued on next page*

*Figure 5 continued*

genes involved in the cellular response to type II interferon (Gene Ontology GO:0071346). (**E**) Shown is a Dotplot visualizing the normalized counts of the type I IFN signature Socs3 from all RNAseq conditions. Clustering was used to (**F**) Significant gene sets from *Med16* KO cells that were uniquely regulated from the RNAseq dataset were analyzed by gene set enrichment analysis (GSEA) then subjected to Leading Edge analysis, which identified a significant enrichment of the cellular responses to type I interferons (normalized enrichment score 2.81, FDR < 0.01). (**G**) Shown is a heatmap demonstrating the relative expression of the type I interferon signature identified in IFN $\gamma$ -stimualted *Med16* KO macrophages from the RNAseq analysis. (**H**) GFP+ NTC cells were mixed equally with GFP- NTC or GFP- *Med16* KO cells. The following day cells were stimulated with 6.25 ng/ml IFN $\gamma$ and 24 hours later MHCII expression was quantified on each cell type. (**Top**) Shown is a representative flow cytometry plot to identify the cells of interest and MHCII expression. (**Bottom**) the % MHCII positive was calculated for cells in each population in each well. Lines link samples that were within the same well. These data are from three biological replicates and represent three independent experiments. **p < 0.01 by two-tailed t-test. (**I**) Shown is a heatmap demonstrating the relative expression of unique differentially expressed genes from the *Gsk3b* KO in the presence (Top) and absence (Bottom) of IFN $\gamma$ -stimulation. (**J**) These differentially expressed genes were used in GSEA to identify Leading Edge networks that are specific to *Gsk3b* KO cells. (Top) Shown is the leading-edge analysis of the UPAR pathway that was identified from IFN $\gamma$ -stimulated *Gsk3b* KO cells. (Bottom) Shown is the leading-edge analysis of the Granulocyte chemotaxis pathway that was identified as differentially regulated in resting *Gsk3b* KO cells.

The online version of this article includes the following figure supplement(s) for figure 5:

**Source data 1.** RNAseq Analysis.

**Figure supplement 1.** Transcriptomic analysis of MED16 and GSK3 $\beta$ reveals mechanisms of IFN $\gamma$ -mediated control.

to NTC due to the muted induction of *Ciita, H2-Ab1* and *Cd74*. This analysis also identified signatures of *Il10, Stat3,* and *Ppar$\gamma$* activation that included *Socs3* induction and *Ptgs2* downregulation (***Figure 5E*** and ***Figure 5—figure supplement 1A*** and S5B). As the DEG analysis relied on a stringent threshold that filtered the great majority of the transcriptome from analysis, we sought to incorporate a more comprehensive analysis capable of capturing genes with more modest effects based on pathway enrichment. To this end, we performed gene set enrichment analysis (GSEA) using a ranked gene list derived from the differential gene expression analysis (***Subramanian et al., 2005***). Of the ~10,000 gene sets tested, 11 sets were enriched for NTC+ IFN$\gamma$ and 76 for MED16+ IFN$\gamma$ (FDR < 0.1). To reduce pathway redundancy and infer biological relevance from the gene sets, we consolidated the signal into pathway networks (***Figure 5—figure supplement 1C***), and observed a significant enrichment for genes involved in xenobiotic and steroid metabolism, including many cytochrome p450 family members and glutathione transferases. We also observed an elevated type I interferon transcriptional response in *Med16* KO cells stimulated with IFN$\gamma$ that included components of IFN$\alpha$/$\beta$ signal transduction (*Ifnar2*), transcription factors (*Stat2, Irf7*) and antiviral mediators (*Oas2, Ifitm1, Ifitm2, Ifitm3, Ifitm6*) (***Figure 6F and G***). Type I IFN production is described to have varying effects on MHCII expression (***Jayarapu et al., 2009***; ***Kurche et al., 2012***; ***Lu et al., 1995***; ***Simmons et al., 2012***). While some studies indicate type I IFN can enhance MHCII in DCs, other studies in distinct cell types suggest type I IFN blunts IFN$\gamma$-mediated MHCII expression. We reasoned that if increased type I IFN in *Med16* KO cells was blocking MHCII expression the type I IFN would also inhibit MHCII expression in wild type cells in *trans*. To test the hypothesis that *Med16* KO cells produce elevated type I IFN that blocks IFN$\gamma$-mediated MHCII induction we conducted a co-culture experiment. *Med16* KO and GFP expressing NTC macrophages were mixed equally, and the following day stimulated with IFN$\gamma$. The surface expression of MHCII was then quantified by flow cytometry. While *Med16* KO cells were unable to robustly induce MHCII, NTC cells from the same well induced MHCII over 30-fold (***Figure 5H***). These data suggest that the effect of Med16 on IFN$\gamma$-mediated MHCII expression is cell-autonomous. Thus, MED16 is a critical regulator of the overall interferon response in macrophages.

We next examined the regulatory networks that were specifically controlled by GSK3$\beta$. As observed by the initial PCA (***Figure 5A***), the transcriptional landscape of GSK3$\beta$ deficient macrophages was altered in unstimulated cells. We hypothesized that these widespread differences may alter cellular physiology and explain, in part, the varied responsiveness of *Gsk3b* KO cells to IFN$\gamma$. DEG analysis of unstimulated macrophages identified 284 differentially expressed genes due to *Gsk3b* loss. Functional enrichment by STRING identified three major clusters that included dysregulation of chemokines, cell surface receptors, growth factor signaling, and cellular differentiation (***Figure 5—figure supplement 1D***). GSEA identified a strong enrichment for chemotaxis and extracellular matrix remodeling pathways including several integrin subunits and matrix metalloproteinase members (***Figure 5I and J***). These results suggest that GSK3$\beta$ is an important regulator of both macrophage homeostasis and the response to IFN$\gamma$. Altogether the global transcriptional profiling suggests that while MED16

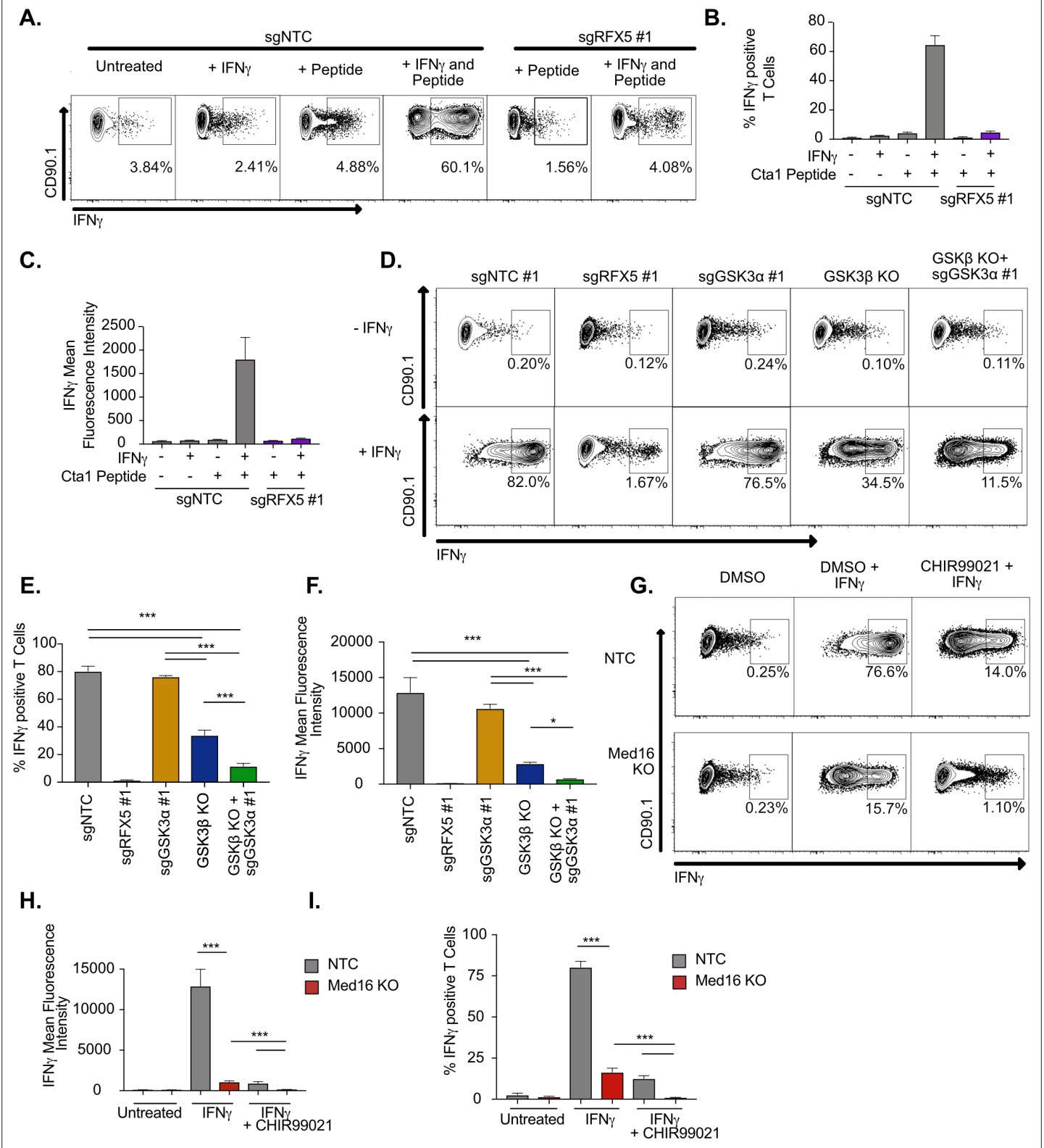

**Figure 6.** IFN $\gamma$ -stimulated macrophages require MED16 or GSK3 to activate CD4[+] T cells. (**A**) Macrophages were left untreated, treated with 10 ng/ml IFN $\gamma$ overnight, 5 μM peptide for 1 hr or both IFN $\gamma$ and peptide as indicated. TCR-transgenic NR1 CD4[+] T cells specific for the peptide Cta1 from *Chlamydia trachomatis* were then added to L3 macrophages of the indicated genotypes at a 1:1 ratio. Four hr after the addition of T cells, NR1 cells were harvested and the number of IFN $\gamma$ -producing CD4[+] T cells was quantified by intracellular staining and flow cytometry. Shown is a representative

*Figure 6 continued on next page*

Figure 6 continued

flow cytometry plot gated on live/CD4$^+$ cells. Gates for IFN$\gamma^+$ T cells were determined using an isotype control antibody. (**B**) The percent of live CD4$^+$ T cells producing IFN$\gamma$ and (**C**) the MFI of IFN$\gamma$ production by live CD4$^+$ T cells was quantified from triplicate samples. These results are representative of three independent experiments. (**D**) L3 cells targeted with the indicated sgRNAs were left untreated or treated overnight with IFN$\gamma$ then pulsed with Cta1 peptide for 1 hr. NR1 cells were then added at a 1:1 ratio and 4 hr later NR1 cells were harvested and the number of IFN$\gamma$-producing CD4$^+$ T cells was quantified by intracellular staining and flow cytometry. Shown is a representative flow cytometry plot gated on live/CD4$^+$ cells. Gates for IFN$\gamma^+$ T cells were determined using an isotype control antibody. (**E**) The percent of live CD4$^+$ T cells producing IFN$\gamma$ and (**F**) the MFI of IFN$\gamma$ production by live CD4$^+$ T cells was quantified from triplicate samples. These results are representative of three independent experiments. (**G**) NTC L3 cells or *Med16* KO cells were left untreated or treated overnight with DMSO, IFN$\gamma$, and DMSO or IFN$\gamma$ and CHIR999021 then pulsed with Cta1 peptide for 1 hour. NR1 cells were then added at a 1:1 ratio and 4 hours after the addition of T cells, NR1 cells were harvested and the number of IFN$\gamma$-producing CD4$^+$ T cells was quantified by intracellular staining and flow cytometry. Shown is a representative flow cytometry plot gated on live/CD4$^+$ cells. Gates for IFN$\gamma^+$ T cells were determined using an isotype control antibody. (**H**) The percent of live CD4$^+$ T cells producing IFN$\gamma$ and (**I**) the MFI of IFN$\gamma$ production by live CD4$^+$ T cells was quantified from triplicate samples. These results are representative of three independent experiments. *** $p < 0.001$, *$p < 0.05$ by one-way ANOVA with a Tukey correction test.

and GSK3$\beta$ are both critical regulators of IFN$\gamma$-mediated MHCII expression, they each control distinct aspects of the macrophage response to IFN$\gamma$.

## Loss of MED16 or GSK3 inhibits macrophage-mediated CD4$^+$ T cell activation

While the data to this point suggested that MED16 and GSK3$\beta$ control the IFN$\gamma$-mediated induction of MHCII, in addition to distinct aspects of the IFN$\gamma$-response, it remained unclear how loss of GSK3$\beta$ or MED16 in macrophages altered the activation of CD4$^+$ T cells. To test this, we optimized an *ex vivo* T cell activation assay with macrophages and TCR-transgenic CD4$^+$ T cells (NR1 cells) that are specific for the *Chlamydia trachomatis* antigen Cta1 (*Roan et al., 2006*). Resting NR1 cells were added to non-targeting control macrophages that were untreated, IFN$\gamma$ stimulated, Cta1 peptide-pulsed, or IFN$\gamma$-stimulated and Cta1 peptide-pulsed. Five hours later, we harvested T cells and used intracellular cytokine staining to identify IFN$\gamma$ producing cells by flow cytometry. Only macrophages that were treated with IFN$\gamma$ and pulsed with Cta1 peptide were capable of stimulating NR1 cells to produce IFN$\gamma$ (*Figure 6A-C*). Additionally, when *Rfx5* deficient macrophages were pulsed with peptide in the presence and absence of IFN$\gamma$, we observed limited IFN$\gamma$ production by NR1 cells in both conditions suggesting this approach is peptide-specific and sensitive to macrophage MHCII surface expression.

We next determined the effectiveness of macrophages lacking GSK3 components to activate CD4$^+$ T cells. Macrophages deficient in *Gsk3a*, *Gsk3b* or both along with NTC and *Rfx5* controls were left untreated or stimulated with IFN$\gamma$ for 16 hours, then all cells were pulsed with Cta1 peptide. Resting NR1 cells were then added and the production of IFN$\gamma$ by NR1 cells from each condition was quantified by flow cytometry five hours later. In agreement with our findings on MHCII expression, loss of *Gsk3a* did not inhibit the production of IFN$\gamma$ by NR1 cells (*Figure 6D-F*). In contrast, *Gsk3b* KO cells reduced the number of IFN$\gamma^+$ NR1 cells over twofold and reduced the mean fluorescence intensity of IFN$\gamma$ production over 4-fold. Furthermore, macrophages deficient in *Gsk3a* and *Gsk3b* were almost entirely blocked in their ability to activate IFN$\gamma$ production by NR1 cells. Thus, macrophages deficient in GSK3 function are unable to serve as effective antigen-presenting cells to CD4$^+$ T cells.

The *ex vivo* T cell assay was next used to test the effectiveness of *Med16* KO macrophages as APCs. NR1 cells stimulated on IFN$\gamma$-activated *Med16* KO macrophages were reduced in the number of IFN$\gamma^+$ T cells by 10-fold and the fluorescence intensity of IFN$\gamma$ by 100-fold compared to NTC (*Figure 6G-I*). Similar to what we observed with MHCII expression, there was a small yet reproducible induction of IFN$\gamma^+$ NR1 cells incubated with IFN$\gamma$-activated *Med16* KO macrophages. We hypothesized that inhibition of GSK3 and MED16 simultaneously would eliminate all NR1 activation on macrophages. Treatment of *Med16* KO macrophages with CHIR99021 prior to IFN$\gamma$-stimulation and T cell co-incubation, eliminated the remaining IFN$\gamma$ production by NR1 cells seen in the DMSO treated *Med16* KO condition. Altogether these results show that GSK3$\beta$ and MED16 are critical regulators of IFN$\gamma$ mediated antigen presentation in macrophages and their loss prevents the effective activation of CD4$^+$ T cells.

## Discussion

IFNγ-mediated MHCII is required for the effective host response against infections. Here, we used a genome-wide CRISPR library in macrophages to globally examine mechanisms of IFNγ-inducible MHCII expression. The screen correctly identified major regulators of IFNγ-signaling, highlighting the specificity and robustness of the approach. In addition to known regulators, our analysis identified many new positive regulators of MHCII surface expression. While we validated only a subset of these candidates, the high rate of validation suggests many new regulatory mechanisms of IFNγ-inducible MHCII expression in macrophages. While the major pathways identified from the candidates in CRISPR screen were related to IFNγ-signaling, we also identified an important role for other pathways including the mTOR signaling cascade. Within the top 100 candidates of the screen several genes related to metabolism and lysosome function including *Lamtor2* and *Lamtor4* were found. Given the known effects of IFNγ in modulating host metabolism, these results suggest that the metabolic changes following IFNγ-activation of macrophages is critical for key macrophage functions including the surface expression of MHCII (*Siska and Rathmell, 2016*). Future studies will need to dissect the metabolism specific mechanisms that macrophages use to control the IFNγ response, including the regulation of MHCII.

In this study, we focused our followup efforts from validated candidates on genes that might control MHCII transcriptional regulation. We identified MED16 and GSK3β as strong regulators of IFNγ-mediated *Ciita* induction. Using global transcriptomics we found that loss of either *Med16* or *Gsk3b* in macrophages inhibited subsets of IFNγ-mediated genes including MHCII. Importantly, the evidence here strongly supports a model where MED16 and GSK3β control IFNγ-mediated MHCII expression through distinct mechanisms (*Figure 7*). Our results uncover previously unknown regulatory control of CIITA-mediated expression that is biologically important to activate CD4$^+$ T cells.

MED16 is a subunit of the mediator complex that is critical to recruit RNA polymerase II to the transcriptional start site (*Poss et al., 2013*). While the mediator complex can contain over 20 unique subunits and globally regulate gene expression, individual mediator subunits control distinct transcriptional networks by interacting with specific transcription factors (*Poss et al., 2013*; *Conaway and Conaway, 2011*). Our data shows that MED16 is uniquely required among the mediator complex for IFNγ-mediated MHCII expression. While we observed a strong reduction in *Ciita* expression in the absence of Med16, some *Ciita* expression remained driving reduced MHCII expression (*Figure 5—source data 1*). Yet how MED16 controls *Ciita* expression upstream of MHCII remains an open question. One recent study showed that MED16 controls NRF2 related signaling networks that respond to oxidative stress (*Sekine et al., 2016*). A major finding of our MED16 transcriptional analysis was the identification of several metabolic pathways involved in oxidative stress and xenobiotics. Given the previous work that described how oxidative stress and the NRF2 regulator KEAP1 regulated IFNγ-mediated MHCII expression in human melanoma cells, NRF2 regulation and redox dysregulation could explain a possible mechanism for MED16 control of MHCII (*Wijdeven et al., 2018*). Intriguingly, the effect of MED16 loss was negligible on many STAT1 and IRF1 targets, and, in fact, resulted in a type I interferon gene signature. Further experiments found that co-culture of *Med16* KO with NTC cells did not alter MHCII expression in either population suggesting a cell-autonomous effect of *Med*16 KO. Thus, what is driving the type I signature following type II interferon activation remains unknown suggesting a careful balance between regulation of distinct IFN-mediated gene expression signatures.

Previous studies showed that CDK8, a kinase that can associate with the mediator complex, controls a subset of IFNγ-dependent gene transcription (*Bancerek et al., 2013*). However, our results strongly support a model where MED16 acts independently of CDK8. Not only was CDK8 not identified in the initial CRISPR screen, but our transcriptional profiling showed that the major IFNγ-dependent genes controlled by *Cdk8*, *Tap1* and *Irf1*, remain unchanged in *Med16* KO macrophages. Thus, understanding what transcription factors MED16 interacts with in the future will be needed to fully determine the mechanisms of MED16-dependent transcription and its control over *Ciita* and IFNγ-mediated gene expression.

While we hypothesize that MED16 directly controls *Ciita* transcription, GSK3 likely regulates MHCII through signaling networks upstream of transcription initiation. GSK3α and GSK3β are multifunctional kinases that regulate diverse cellular functions including inflammatory and developmental cascades (*Wu and Pan, 2010*). Our studies found that GSK3β and GSK3α coordinate IFNγ-mediated MHCII expression, with GSK3β playing a primary role and GSK3α contributing in the absence of GSK3β. The

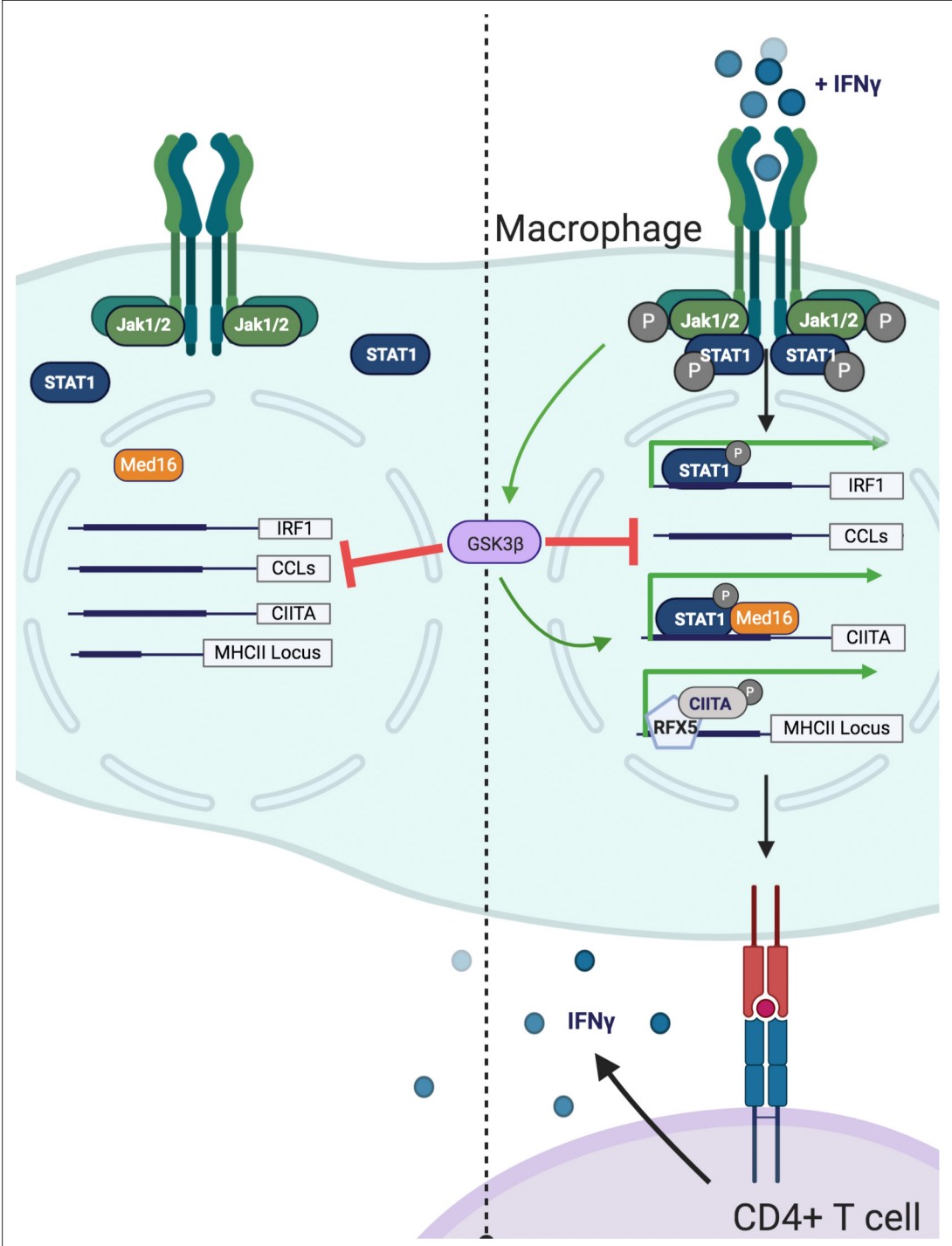

**Figure 7.** Model of GSK3$\beta$− and Med16-mediated control of IFN$\gamma$-activated MHCII expression. Shown is a model of how GSK3$\beta$ and MED16 regulate IFN$\gamma$-mediated MHCII expression. In the absence of IFN$\gamma$ (**Left**) GSK3$\beta$ controls the transcription of many macrophage genes related to inflammation such as CCLs. In contrast, *Med16* KO cells shows minimal transcriptional changes in resting macrophages. Additionally, IFN$\gamma$-mediated gene expression is low. Following the activation of macrophages with IFN$\gamma$ (**Right**), STAT1 becomes phosphorylated and translocates to the nucleus to drive gene transcription. The IFN$\gamma$-mediated induction of *Irf1* does not require either GSK3$\beta$ or MED16. While GSK3$\beta$ continues to negatively regulate inflammatory genes like CCLs it also positively regulates the transcriptional activation of *Ciita* following IFN$\gamma$-activation. Through a parallel but distinct mechanism, IFN$\gamma$-mediated induction of *Ciita* also requires MED16 function. The expression of *Ciita* then recruits other transcription factors such as RFX5 to the MHCII locus where it induces the expression of MHCII, which allows for the activation of CD4+ T cells. Figure created using Biorender.

mechanism of this compensation, however, appears independent of protein abundance or phosphorylation and remains unclear. One possibility is that GSK3β outcompetes GSK3α for substrates related to MHCII expression but testing this hypothesis will require further biochemical studies. Thus, GSK3α and GSK3β are partially redundant in their control of IFNγ-mediated MHCII expression highlighting the interlinked regulation of MHCII.

Because GSK3α/β control many pathways, careful work is needed to determine which networks upstream and downstream of GSK3α/β are responsible for controlling *Ciita* expression. Previous studies suggested that GSK3 controls IFNγ mediated STAT3 activation, LPS-mediated nitric oxide production, and IRF1 transcriptional activity but our transcriptional results clearly show these do not explain the requirement for GSK3-dependent MHCII expression (*Beurel and Jope, 2008*; *Huang et al., 2009*; *Garvin et al., 2019*). Work in human monocyte-derived macrophages showed previously that IFNγ primed macrophages activate mTORC1 resulting in blunted TLR2 responses opposite of the results from the MHCII genetic screen (*Su et al., 2015*). Given GSK3 was previously shown to be modified by mTORC1, we directly examined how mTORC1 modulates IFNγ-mediated responses in the presence and absence of functional GSK3α/β (*Turnquist et al., 2010*). Our study provides new evidence that mTORC1 differentially controls the expression of distinct IFNγ-inducible genes. Blocking mTORC1 activation enhanced IFNγ-mediated PD-L1 surface expression in line with observations in human cells (*Su et al., 2015*). In contrast, mTOR activity was required for robust IFNγ-mediated MHCII expression, in agreement with the bioinformatic analysis from our screen. We also observed that mTORC1 inhibition further diminished MHCII expression in *Gsk3b* KO or CHIR99021 cells suggesting GSK3α/β functions independently of mTOR to control IFNγ-inducible MHCII. Thus, our findings suggest that mTORC1 is both a positive and negative regulator of IFNγ responses that functions independently of GSK3β and Med16 to control MHCII expression. Given mTORC1 is the target of many therapeutics, the mechanisms regulating this differential control of IFNγ-activated pathways will be important to understand.

One additional function of GSK3 is to modulate the activation of the Wnt signaling cascade (*Wu and Pan, 2010*). Inhibition or loss of GSK3 results in the constitutive stabilization of Beta-Catenin and *Tcf* expression. If the constitutive activation of Beta-catenin and Wnt signaling prevents effective *Ciita* expression remains to be determined. Interestingly, another Wnt signaling pathway member *Fzd4* was identified in our screen as required for MHCII expression in our screen, supporting a possible role for Wnt in IFNγ-induced MHCII regulation. It is tempting to speculate that Wnt signaling balances IFNγ-induced activation, resulting in distinct MHCII upregulation between cells with different Wnt activation states. While there is data supporting interactions between Wnt pathways and Type I IFN during viral infections, this has not been explored yet in the context of IFNγ(*Smith et al., 2017*; *Bai et al., 2017*).

GSK3 was recently found to be co-opted by the *Salmonella enterica* serovar Typhimurium effector SteE to skew infected macrophage polarization and allow infection to persist (*Gibbs et al., 2020*; *Panagi et al., 2020*). Our results suggest another possible effect of targeting GSK3 may be the inefficient upregulation of MHCII on Salmonella-infected macrophages in response to IFNγ. While it is known that *Salmonella* and other pathogens including *M. tuberculosis* and *C. trachomatis*, modulate the expression of MHCII, the precise mechanisms underlying many of these virulence tactics remains unclear (*Alix et al., 2020*; *Ankley et al., 2020*). Our screening results provide a framework to test the contribution of each candidate MHCII regulator during infection with pathogens that target MHCII. These directed experiments would allow the rapid identification of possible host-pathogen interactions. It will be important to determine if augmenting specific MHCII pathways identified by our screen overcomes pathogen-mediated inhibition and induces robust MHCII expression to better activate CD4+ T cells and protect against disease using *in vivo* models. Conditional knockout mice were recently developed for GSK3α and Gsk3β and can now be used to specifically ablate *Gsk3b* in macrophages *in vivo* and examine IFNγ responses. However, previous work targeting *Med16* found this knockout is embryonic lethal thus work is underway to develop conditional *Med16* knockout animals to specifically test Med16 function in IFNγresponses to infection *in vivo*.

Beyond infections, our dataset provides an opportunity to examine the importance of newly identified MHCII regulators in other diseases such as tumor progression and autoimmunity. Of course, MHCII is not the only surface marker that is targeted by pathogens and malignancy. Other important molecules including MHCI, CD40, and PD-L1 are induced by IFNγ stimulation and are targeted in

different disease states (*Garcia-Diaz et al., 2017*; *Mandai et al., 2016*; *Gu et al., 2012*; *Zhou, 2009*). Employing our screening pipeline for a range of surface markers will identify regulatory pathways that are shared and unique at high resolution and provide insights into targeting these pathways therapeutically. Taken together, the tools and methods developed here identified new regulators of IFNγ-inducible MHCII that will illuminate the underlying biology of the host immune response.

# Materials and methods

## Key resources table

| Reagent type (species) or resource | Designation | Source or reference | Identifiers | Additional information |
|---|---|---|---|---|
| Cell line (*Mus musculus*) | L3-Cas9+ | This paper | | Primary BMDMs immortalized with J2 virus were transduced with Cas9 and single cell cloned. |
| Cell line (*Mus musculus*) | Med16 KO in L3-Cas9+ | This paper | | L3-Cas9+ cells were transduced with Med16 sgRNAs and single cell cloned |
| Cell line (*Mus musculus*) | GSK3β KO in L3-Cas9+ | This paper | | L3-Cas9+ cells were transduced with GSK3b sgRNAs and single cell cloned |
| Cell line (*Mus musculus*) | Cas9+ C57BL/6 J Estradiol-inducible HoxB8 Progenitors | Kiritsy et al (co-submitted) | | Myeloid progenitors from Jackson stock 026179 were immortalized with HoxB8 retrovirus and maintained with 10 uM estradiol |
| Strain, strain background (*Mus musculus*) | C57BL/6 J | Jackson Laboratories | Stock 000664 | |
| Strain, strain background (*Mus musculus*) | NR1 TCR-transgenic mice | *Roan et al., 2006* | | Mouse strain generated and maintained by Michael Starnbach |
| Recombinant DNA reagent | sgOpti | Addgene PMID:27708057 | RRID:85,681 | |
| Recombinant DNA reagent | sgOpti with blasticidin and zeocyin selection | This Paper | | sgOpti (RRID 85681) was modified with bacterial selection replaced with Zeocyin and mammalian selection replaced with Blasticidin |
| Recombinant DNA reagent | sgOpti with hygromycin and kanamycin selection | This Paper | | sgOpti (RRID 85681) was modified with bacterial selection replaced with Kanamycin and mammalian selection replaced with Hygromycin |
| Antibody | MHCII-PE, Clone M5/114.15.2 (rat monoclonal) | Biolegend | RRID:AB_313323 | FC (1:800) |
| Antibody | anti-mouse IFN-γ Antibody (rat monoclonal) | Biolegend | Cat# 505807, RRID:AB_315401 | FC (1:200) |
| Recombinant DNA reagent | Mouse CRISPR KO pooled library (BRIE) | Addgene | RRID73,632 | |
| Recombinant DNA reagent | VSVG | Addgene | RRID:8,454 | |

*Continued on next page*

*Continued*

| Reagent type (species) or resource | Designation | Source or reference | Identifiers | Additional information |
|---|---|---|---|---|
| Recombinant DNA reagent | psPax2 | Addgene | RRID:12,260 | |
| Chemical Compound Drug | CHIR99021 | Sigma-Aldrich | Catalog: SML1046 PubChemID: 329825639 | (Resuspended in DMSO) |
| Chemical Compound Drug | Torin2 | Sigma-Aldrich | Catalog:SML1225 CAS #:1223001-51-1 | (Resuspended in DMSO) |
| Antibody | GSK3 (rabbit monoclonal) | Cell Signaling Technology | Catalog #: 4,337 RRID: AB_10859910 | WB (1:1000) |
| antibody | pGSK3a (rabbit monoclonal) | Cell Signaling Technology | Catalog #: 9,316 RRID:AB_659836 | WB (1:1000) |
| Antibody | Phospho-Stat1 Tyr701 Clone 58D6 (rabbit monoclonal) | Cell Signaling Technology | Cat# 5375, RRID:AB_10860071 | WB (1:1000) |
| Peptide, recombinant protein | Cta1$_{133-152}$ (KGIDPQELWVWKKGMPNWEK) | Genescript | | Peptide identified in *Roan et al., 2006* |
| Antibody | Human anti-CD274, B7-H1, PD-L1, Clone 29E (mouse monoclonal) | Biolegend | Cat# 329713, RRID:AB_10901164 | FC (1:400) |
| Commercial assay or kit | Zombie Aqua Fixable Viability Kit | Biolegend | Catalog #: 423,101 | FC (1:100) |
| peptide, recombinant protein | IL-12 | Peprotech | Catalog #: 210–12 | |
| Antibody | Anti-IL4 Clone: 11B11 (rat monoclonal) | Biolegend | RRID:AB_2750407 | Neutralization (1:500) |
| Peptide, recombinant protein | IL-18 | Biolegend | Catalog #: 767,008 | |
| Peptide, recombinant protein | IL-15 | Peprotech | Catalog #: 500-P173 | |
| Commercial assay or kit | One-step RT PCR Kit | Qiagen | Catalog #: 210,215 | |
| Commercial assay or kit | Trizol | ThermoFisher Scientific | Catalog #: 15596026 | |
| Software algorithm | MAGECK | PMID:25476604 | | |
| Peptide, recombinant protein | Interferon-gamma | Biolegend | Catalog #:575,308 | |
| Commercial assay or kit | MojoSORT – Mouse CD4 Naïve T cell Isolation Kit | Biolegend | Catalog #:480,040 | |
| Commercial assay or kit | MojoSORT Mouse NK Cell isolation Kit | Biolegend | Catalog #: 480,050 | |
| Antibody | Beta-actin (rat monoclonal) | Biolegend | Catalog#: 664,802 RRID:AB_2721349 | WB (1:2000) |
| Antibody | Goat anti-Rabbit HRP (goat polyclonal) | Invitrogen | Catalog#: 31,460 | WB (1:1000) |
| Antibody | Goat anti-Mouse HRP (goat polyclonal) | Invitrogen | Catalog #: 31,430 | WB (1:1000) |
| Sequence-based reagent | All oligonucleotide sequences are contained in *Supplementary file 1*. | This Paper | | All oligonucleotide sequences are contained in *Supplementary file 1*. |

## Mice

C57BL/6J (stock no. 000664) were purchased from The Jackson Laboratory. NR1 mice were a gift of Dr. Michael Starnbach (*Roan et al., 2006*). Mice were housed under specific pathogen-free conditions

and in accordance with the Michigan State University Institutional Animal Care and Use Committee guidelines. All animals used for experiments were 6–12 weeks of age.

## Cell culture

Macrophage cell lines were maintained in Dulbecco's Modified Eagle Medium (DMEM; Hyclone) supplemented with 5% fetal bovine serum (Seradigm). Cells were kept in 5% CO2 at 37 C. For HoxB8-conditionally immortalized macrophages, bone marrow from C57BL/6J mice was transduced with retrovirus containing estradiol-inducible HoxB8 then maintained in media containing 10% GM-CSF conditioned supernatants, 10% FBS and 10 µM Beta-Estradiol as previously described (*Wang et al., 2006*). To generate BMDMs cells were washed 3 x in PBS to remove estradiol then plated in 20% L929 condition supernatants and 10% FBS. Eight to 10 days later cells were plated for experiments as described in the figure legends.

## CRISPR screen and analysis

The mouse BRIE knockout CRISPR pooled library was a gift of David Root and John Doench (Addgene #73633) (*Doench et al., 2016*). Using the BRIE library, 4 sgRNAs targeting every coding gene in mice in addition to 1000 non-targeting controls (78,637 sgRNAs total) were packaged into lentivirus using HEK293T cells and transduced in L3 cells at a low multiplicity of infection (MOI <0.3) and selected with puromycin two days after transduction. Sequencing of the input library showed high coverage and distribution of the library (*Figure 1—figure supplement 1*). We next treated the library with IFNγ (10 ng/ml) and 24 hr later the cells were fixed and fluorescence activated cell sorting (FACS) was used to isolate the MHCII^high and MHCII^low bins. Bin size was guided by the observed phenotypes of positive control sgRNAs, such as RFX5, which were tested individually and to ensure sufficient coverage ( > 25 x unselected library) in the sorted populations. Genomic DNA was isolated from sorted populations from two biological replicate experiments using Qiagen DNeasy kits. Amplification of sgRNAs by PCR was performed as previously described using Illumina compatible primers from IDT (*Doench et al., 2016*), and amplicons were sequenced on an Illumina NextSeq500.

Sequence reads were first trimmed to remove any adapter sequence and to adjust for p5 primer stagger. We used bowtie two via MAGeCK to map reads to the sgRNA library index without allowing for any mismatch. Subsequent sgRNA counts were median normalized to control sgRNAs in MAGeCK to account for variable sequencing depth. Control sgRNAs were defined as non-targeting controls as well as genes not-transcribed in our macrophage cell line as determined empirically by RNA-seq (*Figure 5—source data 1*). To test for sgRNA and gene enrichment, we used the 'test' command in MAGeCK to compare the distribution of sgRNAs in the MHCII^high and MHCII^low bins. Notably, we included the input libraries in the count analysis in order to use the distribution of sgRNAs in the unselected library for the variance estimation in MAGeCK. sgRNA cloning sgOpti was a gift from Eric Lander & David Sabatini (Addgene plasmid #85681) (*Fulco et al., 2016*). Individual sgRNAs were cloned as previously described (*Shalem et al., 2014*). Briefly, annealed oligos containing the sgRNA targeting sequence were phosphorylated and cloned into a dephosphorylated and BsmBI (New England Biolabs) digested SgOpti (Addgene#85681) which contains a modified sgRNA scaffold for improved sgRNA-Cas9 complexing. A detailed cloning protocol is available in supplementary methods. To facilitate rapid and efficient generation of sgRNA plasmids with different selectable markers, we further modified the SgOpti vector such that the mammalian selectable marker was linked with a distinct bacterial selection. Subsequent generation of SgOpti-Blasticidin-Zeocin (BZ), SgOpti-Hygromycin-Kanamycin (HK), and SgOpti-G418-Hygromycin (GH) allowed for pooled cloning in which a given sgRNA was ligated into a mixture of BsmBI-digested plasmids. Successful transformants for each of the plasmids were selected by plating on ampicillin (SgOpti), zeocin (BZ), kanamycin (HK), or hygromycin (GH) in parallel. In effect, this reduced the cloning burden 4 x and provided flexibility with selectable markers to generate near-complete editing in polyclonal cells and/or make double knockouts.

## Flow cytometry

Cells were harvested at the indicated times post-IFNγstimulation by scraping to ensure intact surface proteins. Cells were pelleted and washed with PBS before staining for MHCII. MHCII expression was analyzed on the BD LSRII cytometer or a BioRad S3E cell sorter. All flow cytometry analysis was done in FlowJo V9 or V10 (TreeStar).

## Chemical inhibitors and agonists

CHIR99021 (Sigma) was resuspended in DMSO at 10 mM stock concentration. DMSO was added at the same concentration to the inhibitors as a control. Cells were maintained in 5 % $CO_2$. Cells were stimulated with 6.25 ng/ml of IFNγ (Biolegend) for the indicated times in each figure legend before analysis. Torin2 (Sigma) was resuspended in DMSO and diluted to the concentrations indicated in each experiment. PAM3SK4 (Invivogen) NG-MDP (Invivogen), IFNβ (BEI Resources), and TNF (Peprotech) were resuspending in sterile PBS and added to cells at the indicated concentrations in the figure legends.

## NK cell isolation, activation, and co-culture

Untouched naïve NK cells were isolated from spleen homogenates of C57BL/6 J mice using the MojoSort Mouse NK cell isolation kit (Biolegend). NK cells were grown for 7–10 days in RPMI with 10 % FBS, non-essential amino acids, 50 µM b-mercaptoethanol and 50 nM murine IL-15 (Biolegend). NK cells were then activated for 18 hr by adding 2 nM IL-12 and 20 nM IL-18 to cells. NK cells viability, differentiation, and activation was confirmed prior to experiments by flow cytometry using anti-CD335 and anti- IFNγ antibodies in combination with a viability live/dead stain (biolegend).

## Isolation of knockout cells

Cells transduced with either MED16 or GSK3β sgRNAs were stimulated with IFNγ then stained for MHCII 24 hr later. Cells expressing low MHCII were then sorted using a BioRad S3e cell sorter and plated for expansion. Gene knockouts were confirmed by amplifying the genomic regions encoding either MED16 or GSK3β from each cell population in addition to NTC cells using PCR. PCR products were purified by PCR-cleanup Kit (Qiagen) and sent for Sanger Sequencing (Genewiz). The resultant ABI files were used for TIDE analysis to assess the frequency and size of indels in each population compared to control cells.

## RNA isolation

Macrophages were homogenized in 500 µL of TRIzol reagent (Life Technologies) and incubated for 5 minutes at room temperature. A total of 100 µL of chloroform was added to the homogenate, vortexed, and centrifuged at 12,000 x g for 20 min at 4 C to separate nucleic acids. The clear, RNA containing layer was removed and combined with 500 µL of ethanol. This mixture was placed into a collection tube and protocols provided by the Zymo Research Direct-zol RNA extraction kit were followed. Quantity and purity of the RNA was checked using a NanoDrop and diluted to 5 ng/µL in nuclease-free water.

## RNA-sequencing analysis

To generate RNA for sequencing, macrophages were seeded in 6-well dishes at a density of 1 million cells/well. Cells were stimulated for 18 hr with IFNγ (Peprotech) at a final concentration of 6.25 ng/mL, after which RNA was isolated as described above. RNA quality was assessed by qRT-PCR as described above and by TapeStation (Aligent); the median RIN value was 9.5 with a ranger of 8.6–9.9. A standard library preparation protocol was followed to prepare sequencing libraries on poly-A tailed mRNA using the NEBNext Ultra RNA Library Prep Kit for Illumina. In total, 18 libraries were prepared for dual index paired-end sequencing on a HiSeq 2500 using a high-output kit (Illumina) at an average sequencing depth of 38.6e6 reads per library with >93 % of bases exceeding a quality score of 30. FastQC (v0.11.5) was used to assess the quality of raw data. Cutadapt (v2.9) was used to remove TruSeq adapter sequences with the parameters `--cores` = 15 m 1 a AGATCGGAAGAGCACACGTC TGAACTCCAGTCA -A AGATCGGAAGAGCGTCGTGTAGGGAAAGAGTGT. A transcriptome was prepared with the rsem (v1.3.0) command rsem-prepare-reference using bowtie2 (v2.3.5.1) and the gtf and primary *Mus musculus* genome assembly from ENSEMBL release 99. Trimmed sequencing reads

were aligned and counts quantified using rsem-calculate-expression with standard bowtie2 parameters; fragment size and alignment quality for each sequencing library was assessed by estimating the read start position distribution (RSPD) via `--estimate-rspd`. aBriefly, counts were imported using tximport (v1.16.0) and differential expression was performed with non-targeting control ('NTC') and unstimulated ('Condition A') as reference levels for contrasts. For visualization via PCA, a variance stabilizing transformation was performed in DESeq2. Pathway enrichment utilized R packages gage and fgsea or Ingenuity Pathway Analysis (Qiagen). Gene-set enrichment analysis (GSEA) was performed utilized gene rank lists as calculated from defined comparisons in DeSeq2 and was inclusive of gene sets comprised of 10–500 genes that were compiled and made available by the Bader lab (*Reimand et al., 2019*). Pathway visualization and network construction was performed in CytoScape 3.8 using the apps STRING and EnrichmentMap. Pathway significance thresholds were set at an FDR of 0.1 unless specified otherwise.

## Quantitative real-time PCR

PCR amplification of the RNA was completed using the One-step Syber Green RT-PCR Kit (Qiagen). 25 ng of total RNA was added to a master mix reaction of the provided RT Mix, Syber green, gene specific primers (5 uM of forward and reverse primer), and nuclease-free water. For each biological replicate (triplicate), reactions were conducted in technical duplicates in 96-well plates. PCR product was monitored using the QuantStudio3 (ThermoFisher). The number of cycles needed to reach the threshold of detection (Ct) was determined for all reactions. Relative gene expression was determined using the $2^{-ddCT}$ method. The mean CT of each experimental sample in triplicate was determined. The average mean of glyceraldehyde 3-phosphate dehydrogenase (GAPDH) was subtracted from the experimental sample mean CT for each gene of interest (dCT). The average dCT of the untreated control group was used as a calibrator and subtracted from the dCT of each experimental sample (ddCT). $2^{-ddCT}$ shows the fold change in gene expression of the gene of interest normalized to GAPDH and relative to to untreated control (calibrator).

## Immunoblot analysis

At the indicated times following stimulation, cells were washed with PBS once and lysed in on ice using the following buffer: 1 % Triton X-100, 150 mM NaCl, 5 mM KCl, 2 mM MgCl2, 1 mM EDTA, 0.1 % SDS, 0.5 % DOC, 25 mM Tris-HCl, pH 7.4, with protease and phosphatase inhibitor (Sigma #11873580001 and Sigma P5726). Lysates were further homogenized using a 25 g needle and cleared by centrifugation before quantification (Pierce BCA Protein Assay Kit, 23225). Parallel blots were run with the same samples, 15 µg per well. The following antibodies were used according to the manufacturer's instructions:

> Anti-GSK3a - #4,337 Cell Signaling Technology
> Anti-pGSK3a - #9,316 Cell Signaling Technology
> Anti-pStat1 0 #8,826 Cell Signaling Technology
> Anti-mouse β-Actin Antibody, Biolegend Cat# 66,480
> Goat anti-Rabbit IgG (H + L) Secondary Antibody, HRP, Invitrogen 31,460
> Goat anti-Mouse IgG (H + L) Secondary Antibody, HRP, Invitrogen 31,430

## T cell activation assays

CD4[+] T cells were harvested from the lymph nodes and spleens of naive NR1 mice and enriched with a mouse naïve CD4-negative isolation kit (BioLegend) following the manufacturer's protocol. CD4[+] T cells were cultured in media consisting of RPMI 1640 (Invitrogen), 10 % FCS, l-glutamine, HEPES, 50 µM 2-ME, 50 U/ml penicillin, and 50 mg/ml streptomycin. NR1 cells were activated by coculture with mitomycin-treated splenocytes pulsed with 5 µM Cta1$_{133-152}$ peptide at a stimulator/T cell ratio of 4:1. Th1 polarization was achieved by supplying cultures with 10 ng/ml IL-12 (PeproTech, Rocky Hill, NJ) and 10 µg/ml anti–IL-4 (Biolegend) One week after initial activation resting NR1 cells were co-incubated with untreated or IFNγ-treated macrophages of different genotypes, that were or were not pulsed with Cta1 peptide. Six hours following co-incubation NR1 cells were harvested and stained for intracellular IFNγ (BioLegend) using an intracellular cytokine staining kit (BioLegend) as done previously. Analyzed T cells were identified as live, CD90.1[+] CD4[+] cells.

## Statistical analysis, replicates, grouping, and figures

Statistical analysis was done using Prism Version 7 (GraphPad) as indicated in the figure legends. Data are presented, unless otherwise indicated, as the mean ±the standard deviation. Throughout the manuscript, no explicit power analysis was used, but group size was based on previous studies using similar approaches. Throughout the manuscript biological replicate refers to independent wells or experiments processed at similar times. For RT-PCR experiments technical replicates were used and are defined as repeat measures from the same well. Throughout the manuscript groups were assigned based on genotypes and blinding was not used throughout. Independent personnel completed several key figures to ensure robustness. Figures were created in Prism V7 or were created with BioRender.com.

## Acknowledgements

We thank members of the Sassetti, Abramovitch and Olive labs for critical feedback and input throughout the project. We thank Dr. Robert Abramovitch for critical reading of the manuscript. We thank the flow cytometry core at MSU and UMMS for their help in all experiments requiring flow cytometry. We would also like to thank Dr. Michael Starnbach for the gift of the NR1 mice. This work was supported by startup funding to AJO provided by Michigan State University, support from the Arnold O Beckman Postdoctoral fellowship to AJO and grants from the NIH (AI146504, AI132130) DOD (W81XWH2010147) and USDA (NIFA HATCH 1019371).

## Additional information

### Funding

| Funder | Grant reference number | Author |
| --- | --- | --- |
| National Institutes of Health | AI146504 | Andrew J Olive |
| U.S. Department of Agriculture | NIFA HATCH 1019371 | Andrew J Olive |
| National Institutes of Health | AI132130 | Michael C Kiritsy |

The funders had no role in study design, data collection and interpretation, or the decision to submit the work for publication.

### Author contributions

Michael C Kiritsy, Conceptualization, Data curation, Formal analysis, Investigation, Methodology, Validation, Visualization, Writing - original draft, Writing – review and editing; Laurisa M Ankley, Conceptualization, Formal analysis, Investigation, Methodology, Validation, Visualization, Writing - original draft, Writing – review and editing; Justin Trombley, Investigation, Methodology, Writing – review and editing; Gabrielle P Huizinga, Formal analysis, Investigation, Methodology, Writing – review and editing; Audrey E Lord, Investigation, Methodology; Pontus Orning, Roland Elling, Conceptualization, Investigation, Methodology, Writing – review and editing; Katherine A Fitzgerald, Conceptualization, Supervision, Writing – review and editing; Andrew J Olive, Conceptualization, Data curation, Formal analysis, Funding acquisition, Investigation, Methodology, Supervision, Validation, Visualization, Writing - original draft, Writing – review and editing

### Author ORCIDs

Michael C Kiritsy http://orcid.org/0000-0001-8364-8088
Pontus Orning http://orcid.org/0000-0002-6177-6916
Andrew J Olive http://orcid.org/0000-0003-3441-3113

### Ethics

This study was performed in strict accordance with the recommendations in the Guide for the Care and Use of Laboratory Animals of the National Institutes of Health. All of the animals were

handled according to approved institutional animal care and use committee (IACUC) protocols (PROTO201800057) of Michigan State University.

### Decision letter and Author response
Decision letter https://doi.org/10.7554/eLife.65110.sa1
Author response https://doi.org/10.7554/eLife.65110.sa2

## Additional files

### Supplementary files
• Supplementary file 1. Oligonucleotides used in the study.
• Transparent reporting form

### Data availability
Raw sequencing data in FASTQ and processed formats is available for download from NCBI Gene Expression Omnibus (GEO) under accession number GSE162463 (CRISPR Screen) and GSE162464 (RNA sequencing).

The following dataset was generated:

| Author(s) | Year | Dataset title | Dataset URL | Database and Identifier |
|---|---|---|---|---|
| Kiritsy MC, Sassetti CM, Olive AJ | 2020 | Mitochondrial respiration contributes to the interferon gamma response in antigen presenting cells | https://www.ncbi.nlm.nih.gov/geo/query/acc.cgi?acc=GSE162463 | NCBI Gene Expression Omnibus, GSE162463 |
| Kiritsy MC, Ankley LM, Olive AJ | 2020 | A genetic screen in macrophages identifies new regulators of IFNg-inducible MHCII that contribute to T cell activation | https://www.ncbi.nlm.nih.gov/geo/query/acc.cgi?acc=GSE162464 | NCBI Gene Expression Omnibus, GSE162464 |

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
