## [Editor Report]

In this study, Olive and colleagues used a genetic screen to identify new regulators underpinning the ability of the cytokine IFNγ to upregulate MHC class II molecules, of relevance to our understanding of how macrophages are activated by IFNγ to confer host defense during microbial infection. They identified the signaling protein GSK3β, and MED16, a subunit of the Mediator complex previously implicated in gene induction.

---

## [Decision Letter]

**Decision letter after peer review:**

Thank you for submitting your article "A genetic screen in macrophages identifies new regulators of IFNγ-inducible MHCII that contribute to T cell activation" for consideration by *eLife*. Your article has been reviewed by 3 peer reviewers, one of whom is a member of our Board of Reviewing Editors, and the evaluation has been overseen by Wendy Garrett as the Senior Editor. The following individuals involved in review of your submission have agreed to reveal their identity: Maziar Divangahi (Reviewer #2); Ivan Zanoni (Reviewer #3).

Essential revisions:

1) Experimental treatment with IFNγ may not be physiological. In key experiments, authors should try co-culture with activated NK cells +/- IFNγ neutralization. A dose and time response curve of IFNγ treatment may be valuable in key experiments.

2) Comparison to cells not stimulated with IFNγ is needed in key experiments. Comparison to WT cells is needed in Figure 5 A,B.

3) Stimulation with Type I IFN and other PAMPs in key experiments, as comparison to the effects of IFNγ and to broaden the relevance of their findings.

4) More insight into how IFNγ signaling interfaces with GSK3 and MED16 is needed (e.g. role of mTORC1 pathway in regulating GSK3).

5) Can the authors extend their data to an in vivo setting?

6) Can the authors clarify the relative roles of GSK3a and GSK3b? For example, how do the authors explain the lack of a robust phenotype in Figure 3B-F?

*Reviewer #1:*

The authors are encouraged to dig deeper into their biological system. Does IFNγ signaling interface with GSK3 or MED16 to regulate their activity, or do other physiological stimuli interface with GSK3 or MED16 to modulate the ability of IFNγ signaling to regulate MHC class II molecule expression? How do GSK3 and MED16 modulate specific aspects of the IFNγ-mediated transcriptional response?

*Reviewer #2:*

In this manuscript entitled "A genetic screen in macrophages identifies new regulators of IFNγ-inducible MHCII that 2 contribute to T cell activation" by Kiritsy, Akley et al., the authors provide novel insight into the cellular signalling pathways involved in MHC-II upregulation on macrophages following exposure to IFN-γ. Although the potent ability of IFN-γ to induce MHC-II on macrophages has been known for decades, here the authors employ a new-age CRISPR-Cas9 forward genetic screen to reveal novel players in the pathway. Following validation, they elucidate 15 novel candidates and subsequently extensively validate two candidates experimentally: MED16 and Glycogen 186 synthase kinase 3β (GSK3β). Finally, they confirm these two candidates regulate MHC-II upregulation in response to IFN-γ on macrophages, albeit through different mechanisms, and that this functionally contributes to the feed-forward loop of MHC-II upregulation and antigen presentation to CD4^+^ T-cells for further production of IFN-γ.

As the authors rightly point-out, they only validate two candidates, the results of their forward screen can lead to subsequent investigation of many other candidates, which adds tremendous value to the submitted study and ensures further contributions to the literature. Although this reviewer is not an expert in CRISPR-Cas9 editing and genomics and, thus, cannot intimately validate the appropriateness of controls and experimental design of these portions, the conceptual framework of this study is very novel and the experiments with KO macrophages and co-cultures are rigorous and exciting. The major concern of this reviewer is the functional importance of these candidates outside of the contrived conditions of direct IFN-γ application to primarily immortalized macrophages. Therefore, some additional experiments can considerably enhance the immunological relevance of this manuscript.

1. The vast majority of experiments are conducted using a single-dose of exogenous IFN-γ in macrophages, where MHC-II expression is then assessed at one single timepoint. Although the authors performed some co-culture experiments (Figure 6), but these experiments were insufficient considering the cellular source as well as the physiological relevance of IFN-γ in these culture models are not known. Can the author performed experiments with activated NK cells that produce large amounts of IFN-γ? Macrophage expression of MHC-II can be investigated following co-culture with stimulated NK cells. Importantly, in this model system, the specificity of IFN-γ in the upregulation of MHC-II can be interrogated using inhibitors of the receptor or neutralization of IFN-γ in the cultures.

2. Along to the above point, is the function of MED16 and GSK3β in the induction of MHC-II specific to type II IFN? The authors find a substantial type I IFN signature in the MED16-deficient macrophages following IFN-γ stimulation. As there is considerable overlap between these pathways, can type I IFN (e.g. α or β) induce MHC-II expression in these macrophages? Additionally, the authors can stimulate the macrophages with PAMPs, such as Poly I:C, or endotoxin (LPS) and investigate whether type I IFN provides a compensatory mechanism for MHC-II induction, by blocking again IFN-γ or type I IFN in those conditions.

3. Finally, if the authors extend their findings to an in vivo setting, this will tremendously improve the significant of their study. For instance, adoptive transfer of WT or KO macrophages into an LPS- or Poly I:C-treated mice (CD45.1 versus CD45.2) can determine whether the expression of MHC-II will be increased specifically on those transferred macrophages.

*Reviewer #3:*

Throughout the paper there is a lack of comparison with cells that have not been exposed to IFNγ. Data shown in figure 1A, B clearly demonstrate that some regulators can change the basal levels of MHCII. This confounds the interpretation of several data in two main ways: (i) it is impossible to distinguish the effects of genes that regulate the basal expression of MHCII and the ones whose activity is solely induced by IFNγ; (ii) in several experiments the upregulation of MHCII upon IFNγ is expressed as a ratio with untreated cells (eg. Figure 1E), not taking into account the activity of the gene(s) of interest on the basal levels of MHCII.

This issue could be solved by: (i) always showing the activity of the gene(s) of interest also in untreated cells; (ii) by using other inflammatory stimuli -eg PAMPs- that are well known to increase MHCII expression. This would also extend the findings of the authors to other stimuli beside IFNγ.

The finding that the mTOR pathway (Figure S2E) may be involved in MHCII upregulation is of great interest. As reported by the authors, this pathway may converge on the activation of GSK3b. The authors completely overlooked if IFNγ is inducing mTOR activation that in turns regulate GSK3b. This issue could be easily solved by administering commercially available drugs to inhibit mTOR activation. Notably, it has been previously shown that IFNγ inhibits mTORC1 in human macrophages (Nat Imm 2015, Ivashkiv). How is the activation state of the mTOR pathway in the cell line used by the authors? Is it active or inhibited? How does this link to GSK3b activation?

The last two panels of figure 2 rise some concerns/questions. While MHCII is completely dampened in Rfx5 KO, Med16 KO show complete abrogation of Ciita but only partial decrease of MHCII. How do the authors explain this phenotype? CIITA and RFX5 are supposed to work in the same complex. Can the authors show the protein levels of CIITA in the Med16 KO cells? Is it possible that Ciita RNA is completely down but the protein is still partially present? This point of Med16 functioning is critical and may need additional experimental data.

The relationship between GSK3b and α is another critical point. It should be important to immediately introduce the relationship between the two proteins cause although presented as a "strong" phenotype by the authors, the phenotype in figure 3B, C D and F is clearly (very) partial. The authors should immediately clarify this in the text and prepare the reader to the transition from figure 3 to figure 4. Clearly the role of GSKa in the absence of GSKb is pretty strong (and not "minor" as stated by the authors in line 289). Is it possible to explain the partial phenotype of the GSK3b KO cells reported in Figure 3 by a non-homogeneous population of cells in which some of the cells still express GSK3b and/or overexpress GSK3a and/or other components of the mTOR pathway? A subcloning of the GSK3b KO cell line may be useful. Also, are the levels of GSKa unaltered in GSK3b KO cells or a feed-back loop is initiated that can explain the major role palyed by GSKa in GSKb KO, but not WT, cells?

Figure 5A, B misses a key control: a side-by-side analysis of WT cells treated in the same manner. Based on previous figures I am not sure CHIR99021 treatment leaves any partial upregulation of MHCII, meaning that the phenotype seen when the drug is administered may be total and completely independent of Med16-deficiency.

Does figure 5H/I suggest that IFN-I are produced in response to IFNγ stimulation in Med16 KO cells? The authors could simply measure IFN-I in the supernatant of their cells. If this is not the case, how do they explain the IFN-I signature?

---

## [Author Response]

Essential revisions:1) Experimental treatment with IFNγ may not be physiological. In key experiments, authors should try co-culture with activated NK cells +/- IFNγ neutralization. A dose and time response curve of IFNγ treatment may be valuable in key experiments.

These comments were very helpful to highlight that Med16 and GSK3a/b robustly control MHCII as discovered in our genetic screen. There were two key points raised by the reviewers that we have now addressed:

A. The reviewers were concerned with the possibility that the IFNg concentrations used were not physiological. To address this we optimized an NK cell co-culture assay. These experiments showed that while NTC macrophages robustly induced MHCII, Med16 and GSK3b KO cells in addition to cells treated with the GSK3a/b inhibitor CHIR99021 were inhibited. The induction of MHCII was IFNg -dependent as NK cells deficient in IFNg or antibody blockade of the IFNgR did not induce MHCII. Thus, using physiological levels of IFNg provided by NK cells we confirm the requirement for Med16 and GSK3a/b in macrophages for IFNg-dependent MHCII expression. These data are now included in our revised manuscript in Figures 2G and 3G.

B. The reviewers were also interested in how MHCII dynamics change over time and with differing IFNg concentrations. To address this we first examined MHCII expression over 48 hours of IFNg stimulation in NTC, GSK3b and Med16 KO macrophages. Regardless of the time point examined there was a significant reduction in the mRNA expression of H2-aA and the surface expression of MHCII in the KO cells compared to NTC controls. While there was a slight increase in MHCII expression in KO macrophages at 48 hours compared to 24 hours, it remained significantly lower than NTC controls. This supports the conclusion that loss of GSK3b and Med16 prevents robust IFNg-mediated MHCII expression. We further examined the sensitivity of MHCII inhibition in Med16 and GSK3b KO macrophages to IFNg concentration. IFNg concentrations as low as 1.5 ng/ml, ten-fold lower than is standard, was sufficient to induce MHCII in NTC macrophages but did not alter GSK3b or Med16 KO MHCII expression. Thus, the phenotypes we identified in our genetic screen are robust to changes in IFNg activation time and IFNg dose. These experiments have been included in Figures S3B-D and S4B-D.

2) Comparison to cells not stimulated with IFNγ is needed in key experiments. Comparison to WT cells is needed in Figure 5 A,B.

We thank the reviewers for pointing out that we did not include a comparison of basal MHCII expression in our validation studies. We have now included both the fold change and the basal MHCII expression level for all targeted macrophages. We did not observe any significant changes in MHCII expression in any target macrophage in the absence of IFNg-activation. These data are now presented in Figure S2F. Throughout the remainder of the manuscript we include IFNg untreated conditions in key experiments and observed no change in MHCII between Med16 and GSK3b KO cells such as in Figures 2D-G, 3C-D and 4A-D.

We also thank the reviewers for pointing out the missing control. We have now included this control in the revised manuscript in Figure 4A. As expected CHIR99021 treatment of NTC cells resulted in a similar inhibition of IFNg-mediated MHCII expression compared to treatment of GSK3b KO cells. This shows that the further reduction in MHCII on GSK3b KO cells following CHIR99021 treatment is likely due to inhibition of active GSK3a.

3) Stimulation with Type I IFN and other PAMPs in key experiments, as comparison to the effects of IFNγ and to broaden the relevance of their findings.

We agree that understanding other environments where Med16 and GSK3b regulate MHCII expression in other inflammatory environments is essential to fully understand their function. We tested a range of stimuli on immortalized bone marrow derived macrophages as suggested by the reviewers. We found that no other stimulations resulted in a significant change in MHCII expression (See Figure S1C and S1D in the revised manuscript) while several stimulations drove increased surface PD-L1 expression. These results are in line with previous studies in BMDMs and epithelial cells that found TLR stimulation and Type I IFN activation do not alter MHCII expression and in fact inhibit IFNg-mediated MHCII expression. We further confirmed that co-stimulation with IFNg and LPS or PAM results in reduced MHCII expression. Our findings are consistent with a model that MHCII expression in bone-marrow derived macrophages is strongly dependent on IFNg activation thus, we did not include any other stimulations throughout the manuscript as the dynamic range of changes is limited.

4) More insight into how IFNγ signaling interfaces with GSK3 and MED16 is needed (e.g. role of mTORC1 pathway in regulating GSK3).

We thank the reviewers for encouraging us to understand the role of mTORC1 in the regulation of IFNg responses and of GSK3- and Med16-dependent pathways. The previous Ivashkiv study cited by the reviewer showed that macrophages treated with IFNg for 24 hours have reduced responsiveness to TLR2 activation that is mediated by an inhibition of mTORC1. This study suggests that IFNg inhibits mTORC1 signaling, at least after 24 hours of activation. In contrast to these results, bioinformatic analysis from our genetic screen showed that the mTORC1 pathways are required for IFNg-mediated MHCII expression. To resolve these discrepancies, we directly tested the role of mTORC1 activation in modulating IFNg responses. We first treated immortalized macrophages with the mTOR inhibitor Torin2 and found blocking mTORC1 activation inhibits the robust expression of IFNg-dependent MHCII in agreement with our screen results. However, when we examined a different IFNg-inducible marker PD-L1 we observed increased IFNg-mediated expression when mTORC1 is inhibited. These findings are consistent with the original findings by Ivashkiv et al., and are presented in Figure 4B-E. Thus, the role of mTORC1 in regulating IFNg-mediated MHCII and PD-L1 are distinct. This suggests that the role of mTORC1 in modulating specific IFNg responses is pathway dependent. We have also expanded our discussion on mTORC1 and IFNg on lines 543-562.

Another key question raised by the reviewers is whether IFNg responses mediated by GSK3b or Med16 are directly modulated by mTORC1 activity. A subset of GSK3a/b functions previously were shown to be regulated by AKT/mTOR activation suggesting a possible link between these pathways. We inhibited mTOR using Torin2 in Med16 and GSK3b KO cells in addition to CHIR99021 treated cells. Our findings clearly show dose dependent decreases in the remaining IFNg-inducible MHCII and increases in PD-L1 expression in each condition shown in Figure 4D and 4E. These data strongly support a model where Med16, GSK3a/b and mTOR all modulate distinct pathways that contribute to IFNg-mediated MHCII surface expression. Several previous studies have shown that a subset of GSK3a/b substrates such as β-catenin are not affected by changes in mTOR activation providing a possible explanation for these results that will need to be tested in our future studies (Reviewed in PMID 31884070). Altogether, our deeper investigation into the role of the mTOR pathway during IFNg responses as suggested by the reviewers has furthered our understanding of the regulatory mechanisms modulating IFNg-inducible MHCII in macrophages. We have uncovered exciting new parallel pathways that are required for effective cross-talk between the innate and adaptive immunity that we can examine in more detail in the future. We have added an additional figure to the manuscript summarizing these results in Figures 5 and S5. We have also added text to discuss how our findings and those by Ivashkiv suggest complex regulation of IFNg pathways by mTOR on lines 543-562.

5) Can the authors extend their data to an in vivo setting?

We agree that examining how Med16 and GSK3b modulate MHCII in vivo is an important question to fully understand their function. The reviewer asked us to transfer congenically marked macrophages into the lungs of mice that are then treated with LPS or Poly I:C, and examine changes in MHCII. However, given our data presented above that MHCII is not strongly modulated by other stimuli, this experiment is challenging and unlikely to provide informative results. Furthermore, we believe this type of transfer experiment is very similar to the NK-cell co-culture experiments that we now include in the revised manuscript. Rather than mixing IFNg producing cells in a dish, these cells would instead be mixed in the lungs, limiting the additional biological information gleaned while using many animals. Instead, we strongly believe the correct approach to address this question is to generate animals deficient in GSK3b and Med16. Given deletion of both GSK3b and Med16 is embryonic lethal this requires using conditional alleles. We recently acquired GSK3b and GSK3a floxed animals that we are now breeding to generate macrophage specific deletions. We are also designing a Med16 floxed animal with our small animal core. Given the length of time needed to generate these new novel models, we believe these experiments are outside the scope of this manuscript. With the inclusion of NK co-culture data to ensure IFNg levels are physiological, in addition to T-cell co-culture experiments that convincingly show that macrophage changes in MHCII directly alter T cell responses our findings directly relate to cellular interactions seen in vivo. We have also added additional text to the discussion to raise the limitation of our study not using in vivo models and how we will address this in the future on lines 582-589.

6) Can the authors clarify the relative roles of GSK3a and GSK3b? For example, how do the authors explain the lack of a robust phenotype in Figure 3B-F?

We thank the reviewers for pointing out that our original manuscript made the relative roles of GSK3b and GSK3a ambiguous. There were two main concerns, first why is there a moderate phenotype for GSK3b in Figure 3 and second how does GSK3a fit in our model. In the revised manuscript we have made several changes that we believe more clearly articulate our findings with GSK3a and GSK3b.

First, with regard to the phenotypes shown in the original Manuscript, we respectfully disagree that these results are not robust. We isolated a knockout GSK3b clone by cell sorting that was confirmed to be fully edited. By flow cytometry there is a reproducible 3-5-fold reduction in the MHCII MFI in GSK3b KO cells and by RT-PCR almost a 5-10-fold reduction in Ciita and H2-aA mRNA expression. Several of these data are presented on a log10 scale given the magnitude of changes following IFNg activation. The timecourse data added in revision further support that this reduction of MHCII expression is seen over time following IFNg activation. The loss of GSK3b alone is also biologically significant as it reduces the capacity of macrophages to activate T cells effectively, suggesting an important conclusion that smaller changes in MHCII expression can directly modulate T cell function.

However, as the reviewers pointed out correctly partial MHCII expression clearly remains in GSK3b KO cells that we believe is mediated by GSK3a compensation. This was not clearly explained in our initial submission. In our revised manuscript we have reordered figures related to GSK3b and GSK3a to introduce GSK3a and the overlap in functions earlier in the results as requested by the reviewer (New Figure 3). This enabled a clearer discussion on the roles of GSK3a and GSK3b throughout the manuscript. We also include new data probing how GSK3a controls MHCII only in the absence of GSK3b to better understand underlying mechanisms. We found that the protein expression levels of GSK3a remain unchanged in GSK3b knockouts, and the phosphorylation of GSK3a also remains unchanged shown in figure 3H. These data suggest a more complicated mechanism used by GSK3a to provide redundant control in GSK3b KO cells. One possibility is that GSK3b outcompetes GSK3a for substrates related to Ciita expression preventing any major regulatory role for GSK3a when GSK3b is present. However, in the absence of GSK3b, these substrates are freed allowing GSK3a to regulate their function. Testing these hypotheses are biochemically intensive experiments that we are excited to pursue in the future. We have added this possibility to the discussion on lines 535-542 where we also more formally discuss the roles of GSK3b and GSK3a to clarify their functions.

Reviewer #1:The authors are encouraged to dig deeper into their biological system. Does IFNγ signaling interface with GSK3 or MED16 to regulate their activity, or do other physiological stimuli interface with GSK3 or MED16 to modulate the ability of IFNγ signaling to regulate MHC class II molecule expression? How do GSK3 and MED16 modulate specific aspects of the IFNγ-mediated transcriptional response?

We thank the reviewer for their recommendations. Our revised manuscript has added additional data to support our models and examine further mechanisms of this pathway. As discussed above new findings have been added related to : (A) the interplay between GSK3a and GSK3b (B) the role of mTOR in regulating Med16/GSK3 pathways (C) NK cell co-culture experiments to understand physiological activation (D) examination of other stimuli that may modulate MHCII function. With these these additions, combined with the data from the original manuscript including a genome-wide genetic screen, RNAseq analysis and T cell interaction studies our manuscript tells a convincing and complete story about regulators of MHCII expression.

Reviewer #2:In this manuscript entitled "A genetic screen in macrophages identifies new regulators of IFNγ-inducible MHCII that 2 contribute to T cell activation" by Kiritsy, Akley et al., the authors provide novel insight into the cellular signalling pathways involved in MHC-II upregulation on macrophages following exposure to IFN-γ. Although the potent ability of IFN-γ to induce MHC-II on macrophages has been known for decades, here the authors employ a new-age CRISPR-Cas9 forward genetic screen to reveal novel players in the pathway. Following validation, they elucidate 15 novel candidates and subsequently extensively validate two candidates experimentally: MED16 and Glycogen 186 synthase kinase 3β (GSK3β). Finally, they confirm these two candidates regulate MHC-II upregulation in response to IFN-γ on macrophages, albeit through different mechanisms, and that this functionally contributes to the feed-forward loop of MHC-II upregulation and antigen presentation to CD4^+^ T-cells for further production of IFN-γ.As the authors rightly point-out, they only validate two candidates, the results of their forward screen can lead to subsequent investigation of many other candidates, which adds tremendous value to the submitted study and ensures further contributions to the literature. Although this reviewer is not an expert in CRISPR-Cas9 editing and genomics and, thus, cannot intimately validate the appropriateness of controls and experimental design of these portions, the conceptual framework of this study is very novel and the experiments with KO macrophages and co-cultures are rigorous and exciting. The major concern of this reviewer is the functional importance of these candidates outside of the contrived conditions of direct IFN-γ application to primarily immortalized macrophages. Therefore, some additional experiments can considerably enhance the immunological relevance of this manuscript.1. The vast majority of experiments are conducted using a single-dose of exogenous IFN-γ in macrophages, where MHC-II expression is then assessed at one single timepoint. Although the authors performed some co-culture experiments (Figure 6), but these experiments were insufficient considering the cellular source as well as the physiological relevance of IFN-γ in these culture models are not known. Can the author performed experiments with activated NK cells that produce large amounts of IFN-γ? Macrophage expression of MHC-II can be investigated following co-culture with stimulated NK cells. Importantly, in this model system, the specificity of IFN-γ in the upregulation of MHC-II can be interrogated using inhibitors of the receptor or neutralization of IFN-γ in the cultures.

We thank the reviewer for recommending these experiments. As discussed above, we have completed these experiments and include them in the revised manuscript.

2. Along to the above point, is the function of MED16 and GSK3β in the induction of MHC-II specific to type II IFN? The authors find a substantial type I IFN signature in the MED16-deficient macrophages following IFN-γ stimulation. As there is considerable overlap between these pathways, can type I IFN (e.g. α or β) induce MHC-II expression in these macrophages? Additionally, the authors can stimulate the macrophages with PAMPs, such as Poly I:C, or endotoxin (LPS) and investigate whether type I IFN provides a compensatory mechanism for MHC-II induction, by blocking again IFN-γ or type I IFN in those conditions.

These are great questions from the reviewer that are addressed in our above responses. We found no other stimulations that significantly alter the expression of MHCII on macrophages. With regard to type I IFN in Med16 KO cells we do not currently understand how this signature is occurring mechanistically. Previous studies suggest differential roles of type I IFN on MHCII dependent on cell type. While type I IFN can modulate MHCII in DCs, our results in macrophages (Figure S1C/D) and other previous studies suggest there may be cell type specificity to the effect of Type I IFN on MHCII expression. For example, Type I IFN has been shown to blunt IFNg-mediated MHCII surface expression in a variety of contexts (PMID: 19667042 and 7595221). Thus, it was possible that Med16 KO cells drive increased Type I IFN that inhibits IFNg-mediated MHCII expression. To test this possibility we co-cultured wild type and Med16 KO cells in the same well and treated with IFNg reasoning that if increased Type I IFN was inhibiting Med16 KO this should inhibit NTC cells in trans. However, our results clearly showed that Med16 KO inhibition of MHCII is cell-autonomous. These new results are now presented in Figure 5H. As the reviewer correctly points out there is significant overlap between the regulons induced by IFNg and Type I IFN. Our current hypothesis is that dysregulated IFNg responses in Med16 KO macrophages results in the loss of regulation over Type I IFN pathways. In future studies we look forward to understanding this delicate balance between Type I and Type II IFN responses in macrophages. This has been further discussed on lines 525-529.

3. Finally, if the authors extend their findings to an in vivo setting, this will tremendously improve the significant of their study. For instance, adoptive transfer of WT or KO macrophages into an LPS- or Poly I:C-treated mice (CD45.1 versus CD45.2) can determine whether the expression of MHC-II will be increased specifically on those transferred macrophages.

We thank the reviewer for this comment that was addressed above.

Reviewer #3:Throughout the paper there is a lack of comparison with cells that have not been exposed to IFNγ. Data shown in figure 1A, B clearly demonstrate that some regulators can change the basal levels of MHCII. This confounds the interpretation of several data in two main ways: (i) it is impossible to distinguish the effects of genes that regulate the basal expression of MHCII and the ones whose activity is solely induced by IFNγ; (ii) in several experiments the upregulation of MHCII upon IFNγ is expressed as a ratio with untreated cells (eg. Figure 1E), not taking into account the activity of the gene(s) of interest on the basal levels of MHCII.This issue could be solved by: (i) always showing the activity of the gene(s) of interest also in untreated cells; (ii) by using other inflammatory stimuli -eg PAMPs- that are well known to increase MHCII expression. This would also extend the findings of the authors to other stimuli beside IFNγ.

We thank the reviewer for these comments. Both issues raised by the reviewer have been addressed in the revised manuscript as described above.

The finding that the mTOR pathway (Figure S2E) may be involved in MHCII upregulation is of great interest. As reported by the authors, this pathway may converge on the activation of GSK3b. The authors completely overlooked if IFNγ is inducing mTOR activation that in turns regulate GSK3b. This issue could be easily solved by administering commercially available drugs to inhibit mTOR activation. Notably, it has been previously shown that IFNγ inhibits mTORC1 in human macrophages (Nat Imm 2015, Ivashkiv). How is the activation state of the mTOR pathway in the cell line used by the authors? Is it active or inhibited? How does this link to GSK3b activation?

As discussed above this was an excellent suggestion by the reviewer that has uncovered a surprising and unexpected differential role of mTORC1 to control IFNg-mediated MHCII or PD-L1 expression. Our inhibitor results suggest that mTOR is activated in our cells during IFNg stimulation and appears to be independent of GSK3b. We are excited to examine the role differential roles of mTOR during IFNg responses in more detail in the future.

The last two panels of figure 2 rise some concerns/questions. While MHCII is completely dampened in Rfx5 KO, Med16 KO show complete abrogation of Ciita but only partial decrease of MHCII. How do the authors explain this phenotype? CIITA and RFX5 are supposed to work in the same complex. Can the authors show the protein levels of CIITA in the Med16 KO cells? Is it possible that Ciita RNA is completely down but the protein is still partially present? This point of Med16 functioning is critical and may need additional experimental data.

We thank the reviewer for asking us to clarify this point. While the reviewer is correct that the RT-PCR data in Figure 2 shows limited Ciita expression, data from replicate experiments in addition to our data from the more sensitive RNAseq do show some Ciita expression (See Figure 5B and 5C) following IFNg activation in Med16 KO. Given that small amounts of Ciita can initiate the assembly of the transcriptional complexes at the MHCII locus, we believe this is responsible for the limited MHCII expression seen. Since none of the macrophages express any Ciita in the absence of IFNg, it is also unlikely that Ciita protein is present without IFNg activation. We have added a line to the text to clarify this point for the readers in the discussion on lines 515-516.

The relationship between GSK3b and α is another critical point. It should be important to immediately introduce the relationship between the two proteins cause although presented as a "strong" phenotype by the authors, the phenotype in figure 3B, C D and F is clearly (very) partial. The authors should immediately clarify this in the text and prepare the reader to the transition from figure 3 to figure 4. Clearly the role of GSKa in the absence of GSKb is pretty strong (and not "minor" as stated by the authors in line 289). Is it possible to explain the partial phenotype of the GSK3b KO cells reported in Figure 3 by a non-homogeneous population of cells in which some of the cells still express GSK3b and/or overexpress GSK3a and/or other components of the mTOR pathway? A subcloning of the GSK3b KO cell line may be useful. Also, are the levels of GSKa unaltered in GSK3b KO cells or a feed-back loop is initiated that can explain the major role palyed by GSKa in GSKb KO, but not WT, cells?

We thank the reviewer for these suggestions. As described above we have addressed each question experimentally and with text to more clearly explain the role of GSK3a/b throughout the manuscript. We believe they have greatly improved the clarity of this section of the revised study.

Figure 5A, B misses a key control: a side-by-side analysis of WT cells treated in the same manner. Based on previous figures I am not sure CHIR99021 treatment leaves any partial upregulation of MHCII, meaning that the phenotype seen when the drug is administered may be total and completely independent of Med16-deficiency.

We thank the reviewer for pointing this out and have added these data to the revised manuscript as discussed above.

Does figure 5H/I suggest that IFN-I are produced in response to IFNγ stimulation in Med16 KO cells? The authors could simply measure IFN-I in the supernatant of their cells. If this is not the case, how do they explain the IFN-I signature?

This is a great question from the reviewer that was also raised by reviewer 2 and is discussed above.